# Genome organization by SATB1 binding to base-unpairing regions (BURs) provides a scaffold for SATB1-regulated gene expression

Yoshinori Kohwi[1,2]*, Xianrong Wong[3†], Mari Grange[1], Thomas Sexton[4], Hunter W Richards[1], Yohko Kitagawa[5‡], Shimon Sakaguchi[5], Ya-Chen Liang[6], Cheng-Ming Chuong[6], Vladimir A Botchkarev[7], Ichiro Taniuchi[8], Karen L Reddy[3], Terumi Kohwi-Shigematsu[1,2]*

[1]Department of Orofacial Sciences, University of California, San Francisco, San Francisco, United States; [2]Lawrence Berkeley National Laboratory, Berkeley, United States; [3]Department of Biological Chemistry, School of Medicine, Johns Hopkins University, Baltimore, United States; [4]Institute of Genetics and Molecular and Cellular Biology (IGBMC), Illkirch, France; [5]Laboratory of Experimental Immunology, Immunology Frontier Research Center, Osaka University, Osaka, Japan; [6]Department of Pathology, University of Southern California, Los Angeles, United States; [7]Department of Dermatology, Boston University School of Medicine, Boston, United States; [8]Laboratory for Transcriptional Regulation, RIKEN Center for Integrative Medical Sciences, Yokohama, Japan

*For correspondence:
YKohwi@lbl.gov (YK);
TKohwi-Shigematsu@lbl.gov
(TK-S)

Present address: †Agency for Science, Technology, and Research (A*STAR) Skin Research Labs (A*SRL), Singapore, Singapore; ‡Center of iPS Research and Applications, Kyoto University, Kyoto, Japan

Competing interest: The authors declare that no competing interests exist.

## eLife Assessment

This is a very **important** study in which the authors have modified ChIP-seq and 4C-seq with a urea step, which drastically changes the pattern of chromatin interactions observed for SATB1, but not other proteins (including CTCF). The study highlights that the urea protocols provide a complementary view of protein-chromatin interactions for some proteins, which can uncover previously hidden, functionally significant layers of chromatin organization. If applied more widely, these protocols may significantly further our understanding of chromatin organization. The study's findings are supported by a wealth of controls, making the evidence **compelling**.

**Abstract** Mammalian genomes are organized by multi-level folding; yet how this organization contributes to cell-type-specific transcription remains unclear. SATB1 forms a nuclear substructure that resists high-salt extraction. SATB1 binds base-unpairing regions (BURs), genomic elements with high unwinding propensities. In mouse thymocytes, we found that SATB1 establishes a two-tiered chromatin organization: one through indirect binding to transcriptionally active DNase 1-accessible chromatin and another by direct binding to BURs in the DNase 1-inaccessible nuclear substructure. Recently published ChIP-seq datasets show SATB1 binding to accessible chromatin at enhancers and CTCF sites, but not to BURs. By employing urea ChIP-seq, which retains only directly bound protein:DNA complexes, we found that BURs, but not CTCF sites, are direct SATB1 binding targets genome-wide. BURs bound to the SATB1 nuclear substructure interact with accessible chromatin, crossing multiple topologically associated domains (TADs). SATB1 is required for these megabase-scale interactions linked to cell-type-specific gene expression. BURs are highly enriched within transcriptionally repressive lamina-associated domains (LADs). Besides these BURs, SATB1 anchors

some BURs (18%) outside LADs near genes in otherwise accessible chromatin to the SATB1 nuclear substructure. Only a subset of total BURs is bound to SATB1, depending on cell type. Notably, despite the mutually exclusive SATB1-binding profiles uncovered by the two ChIP-seq methods, we found most peaks in both profiles are valid and require SATB1. Based on these and previous data, we propose that the SATB1 protein network forms a chromatin scaffold, providing an interface that connects accessible chromatin to a subnuclear architectural structure, thereby facilitating the three-dimensional organization linked to cell-type-specific gene expression.

## Introduction

Three-dimensional (3D) chromatin architecture, formed by chromatin folding at multiple hierarchical levels, is thought to play a fundamental role in genome activity such as gene expression (*Politz et al., 2013*; *van Steensel and Belmont, 2017*; *Schoenfelder and Fraser, 2019*; *Bashkirova and Lomvardas, 2019*; *Göndör and Ohlsson, 2018*; *Loubiere et al., 2020*; *Misteli, 2020*). Whether and how genome folding contributes to gene regulation, however, remains largely unknown. High-resolution chromatin interaction maps, generated by chromosome conformation capture methods such as Hi-C (all-to-all contacts), have revealed chromatin to be partitioned mainly into two types of megabase (Mb)-sized compartments, A and B (*Lieberman-Aiden et al., 2009*). The euchromatic A compartment is gene-rich, transcriptionally active, and enriched in histone marks associated with active chromatin, conferring high chromatin accessibility. On the other hand, the B compartment is gene-poor, transcriptionally repressive, and enriched in repressive histone marks. The largely transcriptionally repressive lamina-associated domains (LADs) are among the B compartment chromatin displaying the typical features of heterochromatin (*Guelen et al., 2008*; *Kind et al., 2015*). LADs have been identified by the lamin B1-mediated adenine methyltransferase identification (DamID) method (*van Steensel et al., 2001*) as genomic regions that are spatially proximal to the nuclear lamina, an intermediate filament meshwork adjacent to the inner nuclear membrane. LADs represent discrete chromatin domains of 10 kb to 10 Mb in size in mouse and human cells. Approximately 1000–1500 LADs have been identified in these cells and they cover greater than 30% of the genome (*Guelen et al., 2008*; *Peric-Hupkes et al., 2010*). By sequestering chromatin to the repressive nuclear compartment, the nuclear lamina is thought to play an important role in genome organization.

At sub-megabase scales, using Hi-C, chromatin domains called 'topological-associated domains' (TADs) were previously detected that are relatively insulated from adjacent TADs by preferentially making intra-domain contacts via looping (*Dixon et al., 2012*; *Nora et al., 2012*; *Sexton et al., 2012*). CTCF and cohesin are the two major architectural proteins required for folding chromatin into loops. The loop extrusion model postulates that cohesin functions in loop extrusion until it encounters the CCCTC-binding factor (CTCF) bound to its target sites in a convergent orientation (*Sanborn et al., 2015*; *Fudenberg et al., 2016*). CTCF- and cohesin-binding sites are enriched at the boundaries of TADs (*Dixon et al., 2012*; *Rao et al., 2014*) and LADs (*Guelen et al., 2008*). Acute depletion of CTCF (*Nora et al., 2017*), cohesin (*Rao et al., 2017*; *Wutz et al., 2017*) or induced removal of cohesin loading factor NIPBL (*Schwarzer et al., 2017*) results in a major loss of TADs. Unexpectedly, the majority of protein-coding genes remained largely unchanged in their expression levels in the absence of TADs. These results challenge the view that TADs provide a unit of gene expression control by insulating enhancer-promoter interactions from neighboring TADs. In the last few years, using super-resolution live-cell imaging, genome interactions were studied at the single cell levels. These studies revealed that chromatin looping formed by CTCF and cohesin is highly dynamic and variable among individual single cells, suggesting that TADs are an emergent property of the cell population average of heterogeneous chromatin folding (*Cattoni et al., 2017*; *Bintu et al., 2018*; *Finn et al., 2019*; *Gabriele et al., 2022*; *Mach et al., 2022*). Therefore, TADs, enriched in contacts, may not represent a stable structural unit of chromatin flanked by two boundary loci stably bound by CTCF and cohesin [reviewed in *Gabriele et al., 2022*; *Chi et al., 2022*; *Chen et al., 2023*]. Super-resolution microscopy detected not only single cohesin-mediated loops but also radially stacked loops forming 'rosette' like structures in individual single cells. These loops were also transient, and a single cell spends a minority of its time in the configuration, and depletion of cohesin led to increased variability in gene expression at single-cell levels (*Hafner et al., 2023*). As such, although research on CTCF and cohesin has greatly advanced our understanding of chromatin loop formation, there is still much to learn about

how and what functional 3D chromatin structure underlies gene activity, especially cell-type-specific gene expression. It is increasingly clear that it is necessary to explore additional nuclear factors that potentially drive chromatin folding. In this paper, we studied the function of a tissue-specific architectural protein SATB1 in genome organization.

SATB1 was originally identified by virtue of its specific binding to genomic sequences called Base Unpairing Regions (BURs). Previous in vitro studies have shown that SATB1 directly binds BURs without requiring additional molecules, such as RNA or proteins (*Kohwi-Shigematsu and Kohwi, 1990*; *Bode et al., 1992*; *Dickinson et al., 1992*). BURs have a strong unwinding property under negative superhelical stress, and BURs are empirically identified using unpaired DNA-specific chemical probes (*Kohwi-Shigematsu and Kohwi, 1990*; *Kohwi-Shigematsu and Kohwi, 1992*) because they lack primary sequence consensus. However, BURs have a unique sequence context: a typical BUR (~200–300 bp) contains a cluster of short sequence segments (25–50 bp/segment) containing exclusively As, Ts, and Cs on one strand (ATC context; *Kohwi-Shigematsu and Kohwi, 1990*; *Dickinson et al., 1992*). BURs are at least 65% AT-rich, but importantly, not all AT-rich sequences are BURs. Importantly, SATB1 binds to BURs in the minor groove of double-stranded DNA, recognizing the altered phosphate backbone structure; it does not bind to single-stranded DNA (*Dickinson et al., 1992*). Disruption of the ATC context by mutagenesis, without affecting the AT content, results in loss of unwinding property of a BUR and SATB1 binding. While BURs are obviously present in the genome across cell types, SATB1 protein is cell-type restricted, most abundantly expressed in thymocytes and among multiple immature progenitor cells of adult tissues (*Fessing et al., 2011*; *Alvarez et al., 2000*; *Zhang et al., 2019*), as well as terminal differentiated postnatal neurons in the cortex and amygdala (*Balamotis et al., 2012*; *Huang et al., 2011*). Located in the nuclear interior, SATB1 has an interconnected cage-like distribution in thymocytes (*Cai et al., 2003*) and a fine spider web-like distribution in neurons (*Balamotis et al., 2012*) forming a unique SATB1-rich subnuclear architecture that is resistant to high-salt extraction. Some BURs have been identified and characterized as in vivo SATB1 binding sites and, when bound to SATB1, are also resistant to salt extraction (*Cai et al., 2003*; *de Belle et al., 1998*). SATB1 has been shown to be essential for loop formation connecting a subset of these BURs and recruiting chromatin remodeling complexes, epigenetic factors, and transcription factors to specific gene loci (*Kohwi-Shigematsu et al., 2013*). Of particular interest, SATB1 regulates expression of hundreds of protein-coding genes in a cell-type-specific manner to enable cells to acquire new phenotypes, such as during T cell activation (*Cai et al., 2006*), development of many adult progenitor cells (*Fessing et al., 2011*; *Balamotis et al., 2012*; *Alvarez et al., 2000*; *Zhang et al., 2019*, *Skowronska-Krawczyk et al., 2014*; *Hao et al., 2015*; *Kakugawa et al., 2017*; *Kitagawa et al., 2017*; *Yasuda et al., 2019*; *Goolam and Zernicka-Goetz, 2017*; *Satoh et al., 2013*), and cancer metastasis (*Kohwi-Shigematsu et al., 2013*; *Han et al., 2008*; *Ordinario et al., 2012*; *Frömberg et al., 2018*). For example, by directing lineage-specific transcriptional programs, SATB1 plays an essential role in the development of CD4[+] T cells, CD8[+] T cells, NKT cells, and Foxp3[+] regulatory T cells (Treg) in the thymus (*Kakugawa et al., 2017*; *Kitagawa et al., 2017*) as well as in promoting the pathogenic effector program of tissue Th17 cells in autoimmune disease in mice (*Yasuda et al., 2019*). These activities have been shown to mediate co-induction of interleukin genes in T helper cells (*Cai et al., 2006*) and regulate ~1000 genes in cancer to promote metastasis (*Han et al., 2008*). Thus, SATB1 has long been thought to play a role in regulating genome architecture and in gene expression.

Although BURs have been identified to be specific and direct binding target sequences of SATB1, mysteriously, the genome-wide SATB1 binding profiles obtained by the widely used, standard chromatin immunoprecipitation-deep sequencing (ChIP-seq) do not detect SATB1 interaction with BURs. Instead, SATB1 has been mapped mainly to enhancers and to a region that would become an active super-enhancer upon differentiation (*Hao et al., 2015*; *Kakugawa et al., 2017*; *Kitagawa et al., 2017*). In line with this, recent studies on chromatin looping with Hi-C also show that SATB1 mediates interactions involving a subset of enhancers and promoters for cell identity gene expression and that SATB1 binding sites coincide with many CTCF-bound loci (*Feng et al., 2022*; *Zelenka et al., 2022*; *Wang et al., 2023*). These results align with SATB1's role in assembling transcription complexes at regulatory regions. However, they are also puzzling, considering that BURs have been identified as the direct binding targets of SATB1 by in vitro binding assays. We therefore hypothesized that SATB1-BUR interactions are somehow undetectable by the standard ChIP-seq approach and CUT&Tag methods,

and if identified, would provide additional mechanisms for chromatin looping events in vivo that link to gene expression.

ChIP-seq has been a gold standard for profiling chromatin-protein interactions. However, standard ChIP-seq approaches have several drawbacks. First, chromatin structure can introduce biases into ChIP-seq analysis. Fragmentation of crosslinked chromatin in whole cells/nuclei can result in biased capture of transcriptionally active, highly accessible 'open' chromatin and generation of false-positive 'phantom' peaks (*Jain et al., 2015*; *Park et al., 2013*; *Teytelman et al., 2013*; *Meyer and Liu, 2014*; *Baumgarten and Bryant, 2022*). Second, it fails to distinguish between direct and indirect (piggy-backing) chromatin binding of a protein of interest. Here, we optimized urea-ChIP-seq, a modified ChIP-seq that first isolates stringently purified whole intact genomic DNA that retains only its directly bound proteins from crosslinked cells prior to fragmentation and immunoprecipitation. This captures the entire genome, regardless of its original chromatin accessibility status. This approach has allowed us to identify genomic sites directly bound by SATB1 and revealed BURs to be the primary *direct* SATB1 target sites, previously hidden by standard ChIP-seq. The SATB1-bound BURs are mutually exclusive with enhancer-enriched SATB1-bound sites mapped by standard ChIP-seq. Instead, a majority of BURs are mapped within LADs, and depending on cell type, SATB1 binds in vivo to selective subsets of these BURs. Using the urea 4C-seq (one-to-all contacts), select SATB1-bound BURs were found to interact extensively over the >5.7 Mb gene-rich region within DNase 1 hypersensitive (DNase 1 HS) regions, but not with adjacent gene-poor regions. SATB1 is essential for both megabase-level chromatin interactions and proper expression of multiple genes in this gene-rich region. Collectively, our results suggest that SATB1-mediated genome organization exhibits two distinct features, a 'skeletal framework' where SATB1 binds BURs directly, and a network of gene-rich open chromatin that is tethered to BURs indirectly through SATB1-mediated chromatin looping, which is correlated to gene expression.

## Results

### Development of urea ChIP-seq method to detect genome-wide SATB1-binding profiles

Unlike many nuclear proteins (e.g. transcription and chromatin modifying factors), SATB1 resides in nuclear substructures exhibiting resistance to extraction with high salt or lithium 3,5-diidosalicylate (LIS) (*de Belle et al., 1998*; *Skowronska-Krawczyk et al., 2014*). Consistent with this, after extraction of nuclei with buffer containing 2 M NaCl that removed most of the proteins from DNA, individually cloned SATB1-bound BURs remained anchored to the residual SATB1 protein network (SATB1-rich subnuclear architecture) found in the interior of nuclei (*Cai et al., 2003*). By contrast, in high-salt extracted *Satb1*$^{-/-}$ (SATB1-KO) thymocyte nuclei, these BURs lost their anchored sites and were exclusively found in the distended DNA halos that spread around the residual nuclei, which is illustrated (*Figure 1A*) (*Cai et al., 2003*). Thus, we hypothesized that SATB1-BUR interactions are likely embedded in an insoluble nuclear substructure that could not be detected to date by the widely used standard ChIP-seq methods. It is possible that ChIP-seq experiments have inherent biases arising from chromatin structure such as heterochromatin and transcriptionally active regions (*Gesson et al., 2016*).

To detect BURs as the primary targets of SATB1 genome-wide, we devised a urea ChIP-seq method to purify genomic DNA with only its directly bound proteins. Through this approach, we found that a vast majority of in vivo SATB1-binding sites are BURs in the genome. In this method, cells are first crosslinked with formaldehyde. Then, cells are lysed in a 4% SDS-containing buffer and subjected to ultracentrifugation with 8 M urea. In this manner, crosslinked chromatin is stringently purified by removing all free-floating proteins and proteins bound to DNA indirectly. Therefore, urea ChIP-seq is performed with the genomic DNA retaining only directly bound proteins. It is important to note that without prior formaldehyde crosslinking, urea centrifugation removes 99.9% of chromatin-associated protein from chromatin and sediments unsheared pure genomic DNA at the bottom of the tube (*Kohwi-Shigematsu et al., 1998*). As a denaturant for DNA, RNA, and proteins, high concentrations of urea (e.g. 6–8 M) disrupt the secondary structures of nucleic acids and the three-dimensional structure of proteins (*Das and Mukhopadhyay, 2009*; *Raghunathan et al., 2020*; *Oprzeska-Zingrebe and Smiatek, 2018*). Crucially, purified genomic DNA obtained from 8 M urea ultracentrifugation remains double-stranded and can therefore be readily digested with restriction enzymes this study and

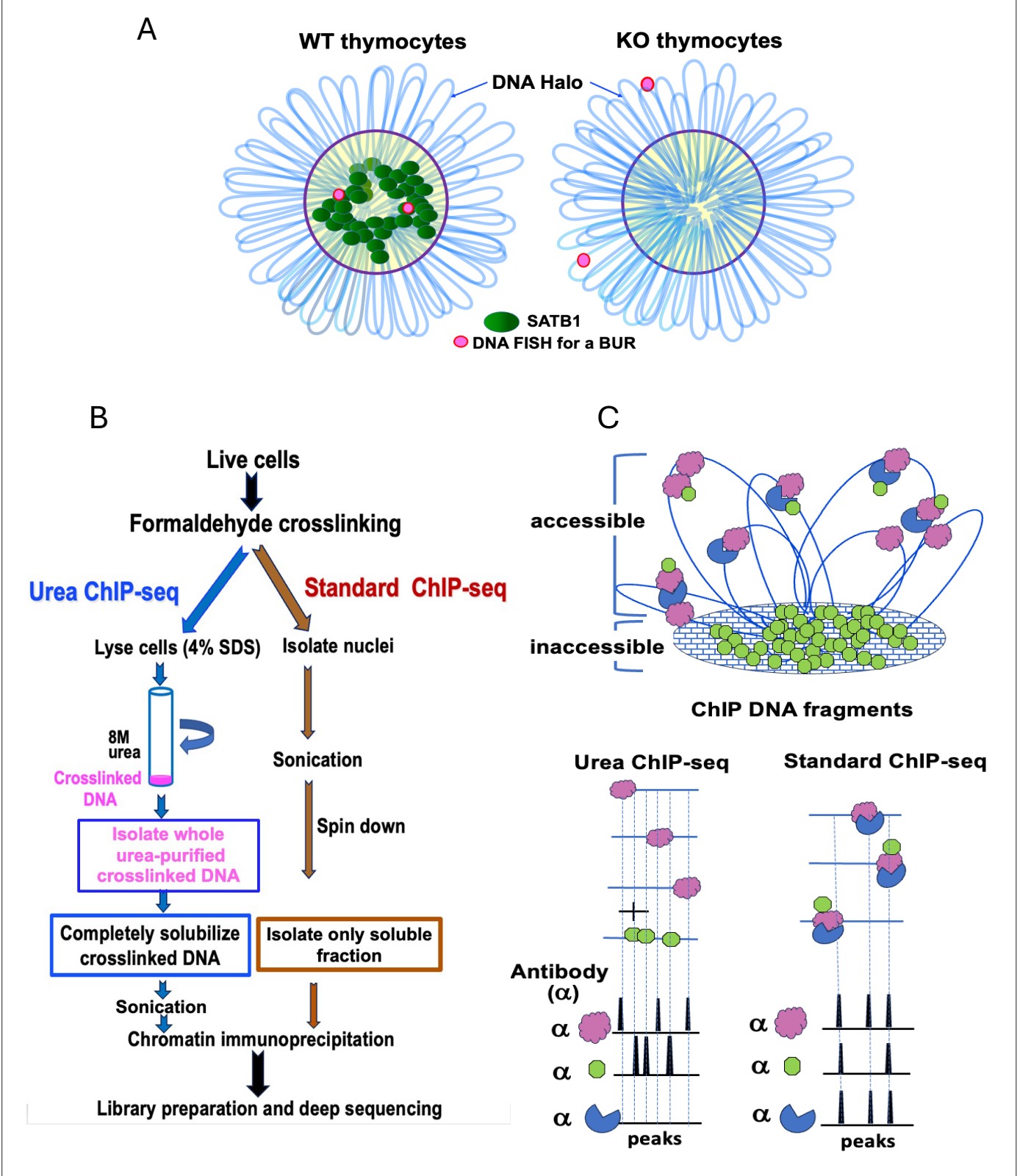

**Figure 1.** Scheme of urea ChIP-seq and difference from standard ChIP-seq. (**A**) An illustration for a BUR anchored to the high-salt extraction resistant SATB1 subnuclear architecture (**Cai et al., 2003**). DNA-FISH signals (red dots) for a cloned SATB1-bound DNA in both alleles are shown in *Satb1*$^{+/+}$(WT) and *Satb1*$^{-/-}$(KO) thymocytes on slides treated with 2 M NaCl solution to produce DNA halos around the nuclei (blue loops). SATB1 and the SATB1-bound DNA remain in the residual nucleus in WT thymocytes after salt treatment. However, in KO thymocytes, the DNA regions normally bound

*Figure 1 continued on next page*

*Figure 1 continued*

by SATB1 are located in the DNA halos after the treatment. (**B**) A comparative overview of the urea ChIP-seq and standard ChIP-seq protocols. In urea ChIP-seq, entire crosslinked chromatin purified by 8 M urea ultracentrifugation is solubilized (see Materials and methods) before chromatin immunoprecipitation. By contrast, in standard ChIP-seq, crosslinked chromatin in whole nuclei is sonicated, centrifuged, and the soluble fraction is isolated for chromatin immunoprecipitation. These critical steps that are different between urea and standard ChIP-seq are highlighted in boxed texts. (**C**) Differences in the profiles of the peaks generated from urea ChIP-seq versus standard ChIP-seq. A diagram is shown for ChIP-seq peaks using antibodies (α) against a DNA direct-binding protein (pink), a DNA indirect-binding protein (blue), and DNA direct-binding protein that resides in the inaccessible chromatin region (green). Note: This diagram illustrates a hypothetical scenario where some of the proteins (green) in the inaccessible chromatin region may also reside in the accessible chromatin binding indirectly to chromatin.

(*Kohwi-Shigematsu et al., 2012*). Additionally, the proteins that remain bound to urea-purified DNA from crosslinked cells can still be detected with the antibodies compatible with standard ChIP protocols. By acting as both a hydrogen bond donor and acceptor, high concentrations of urea destabilize double-stranded DNA by weakening the hydrogen bonds between base pairs, thereby decreasing the melting temperature of DNA (*Kourilsky et al., 1970*; *Kourilsky et al., 1971*). However, even 6–8 M urea alone, without heat treatment, is insufficient to convert double-stranded DNA into single-stranded DNA (*Hegedüs et al., 2009*). Denaturing polyacrylamide electrophoresis for separating DNA oligonucleotides by size typically uses 6–8 M urea. For this, DNA samples need to be heat denatured first to make them single-stranded (*Summer et al., 2009*). Our 8 M urea centrifugation protocol lacks heat treatment and includes prior formaldehyde fixation, thus allowing genomic DNA to remain double-stranded and bound by interacting proteins.

Critically, the urea ChIP-seq method (*Kohwi-Shigematsu et al., 2012*) allows detection of genomic regions solely based on their direct protein binding regardless of their original chromatin status, such as 'accessible' or 'inaccessible' chromatin. To obtain reproducible direct binding profiles of SATB1 at the genome-wide scale, it was necessary to make additional optimizations. The modified urea ChIP-seq protocol allows quantitative solubilization of the unsheared crosslinked genomic DNA fiber after urea ultracentrifugation, while still maintaining intact chromatin-protein interactions during fragmentation. The critical steps that are different between urea and standard ChIP-seq are highlighted (*Figure 1B*). A putative scenario as to how DNA-binding sites of chromatin-associated factors are captured by the two different ChIP-seq methods is illustrated (*Figure 1C*). Urea ChIP-seq allows detection of chromatin-bound proteins regardless of the chromatin's accessibility because the whole genome is purified. In contrast, standard ChIP-seq discards most of the insoluble fraction and retains primarily the soluble fraction for the subsequent chromatin immunoprecipitation step (*Gesson et al., 2016*). This results in samples that are enriched in accessible 'open' chromatin. Because a fraction of SATB1 proteins can be easily extracted from cells with a high salt-containing buffer, whereas other fractions resist extraction, both soluble and insoluble fractions are expected to contain SATB1 (*de Belle et al., 1998*). This could reflect SATB1 proteins being heterogeneous with regard to their mobility. By fluorescence recovery after photobleaching (FRAP) assay in thymocytes, we found one fraction of SATB1 essentially immobile and the other with high mobility (our unpublished results). Due to the lack of specific markers to distinguish these SATB1 proteins, we cannot confirm the exact relationship between fast-mobile SATB1 and immobile SATB1 to the soluble and insoluble fractions, respectively. Comparing the two ChIP-seq protocols, standard ChIP-seq, therefore, captures both directly and indirectly bound proteins for chromatin fragments in the soluble fraction, and their binding profiles cannot be distinguished. In contrast, urea ChIP-seq strongly enriches for directly bound proteins, including those in chromatin regions that are difficult to solubilize. Thus, urea ChIP-seq is uniquely suited for detecting direct DNA binding profiles of SATB1 that resides in the high-salt extraction resistant nuclear substructure.

## Urea ChIP-seq detects BURs as direct binding targets of SATB1 in vivo

To determine whether SATB1 binds to BURs in vivo, it is essential to validate in vivo SATB1-bound sites as BURs. We previously generated a genome-wide BUR reference map by deep sequencing DNA fragments generated from purified genomic DNA bound by recombinant SATB1 protein in vitro (*Kohwi-Shigematsu et al., 2012*). Since SATB1 binds specifically to BURs in vitro, we used SATB1 as a biological probe to identify all potential BURs in the genome. Using this approach, we reproducibly identified approximately 240,000 BURs (q<0.01), establishing a BUR reference map for both mouse

and human genomes. Comparing ChIP-seq peaks for SATB1 with the BUR reference map allows us to assess whether SATB1 binding sites in vivo correspond to BURs.

We performed urea ChIP-seq for SATB1 in thymocytes and compared the results with standard ChIP-seq for this protein and the BUR reference map. Strikingly, urea ChIP-seq produced SATB1 binding profiles that are entirely different from those produced by the standard ChIP-seq method (*Figure 2A*). The genome-wide analysis in mouse thymocytes using deepTools revealed that only 0.92% (an average of two replicates per ChIP-seq method) of peaks in urea SATB1 ChIP-seq coincided with those in standard SATB1 ChIP-seq (*Supplementary files 1 and 2*). In a large genomic region covering a 4.48 Mb region in chromosome 17 (*Figure 2A*), a gene-rich region was found in the valley between BUR-enriched domains (*Figure 2A, track 3*), and standard ChIP-seq detects clusters of SATB1 binding sites in this gene-rich region (*Figure 2A, tracks 1 and 2*). In contrast, urea ChIP-seq detected SATB-binding to the domains surrounding the gene rich region, corresponding to regions heavily enriched in BUR sites (*Figure 2A, tracks 3–5*). In a zoomed-in view, covering a 430 kb region that includes a gene-rich region, SATB1 sites identified from urea ChIP-seq precisely coincided with BUR peaks and are excluded in the standard SATB1 ChIP-seq peaks (*Figure 2B, tracks 1–3*). SATB1 bound sites uncovered from urea ChIP-seq avoided CTCF sites (*Figure 2B, tracks 4 and 5 versus track 6*), H3K27ac-marked sites and DNase 1 hypersensitive (HS) regions (*Figure 2B tracks 7 and 8*), whereas SATB1 sites from standard ChIP-seq primarily co-mapped to these sites (*Figure 2B, tracks 1,2,6–9*). To unequivocally demonstrate that urea ChIP-seq peaks for SATB1 coincide with BURs, we examined these peaks shown in *Figure 2B* at a higher resolution (*Figure 2—figure supplement 1A and B*). All four SATB1 peaks (*Figure 2B, track 4 and 5*) precisely co-map with BURs. For one of these peaks, a typical BUR sequence is shown that contains a cluster of DNA sequence stretches in which either Gs or Cs are lacking from one strand (*Figure 2—figure supplement 1C*). This DNA sequence context is referred to as the ATC sequence context. We confirmed the relationship of SATB1-binding sites by urea ChIP-seq and BURs at the genome-wide level using deepTools. Among SATB1 urea ChIP-seq peaks, an average of 77% of two replicates coincided with BURs identified in our reference maps (*Figure 2C*). The statistical significance of enrichment of BURs compared to a random overlap in either urea ChIP-seq or standard ChIP-seq peaks is p<0.001 or p=1, respectively, using 1000 bootstrapped genomic intervals sampled using the *bootRanges* approach (*Mu et al., 2023*) (*Figure 2C*). Consistent with this, genome-wide distribution profile analyses show that while SATB1 binding profiles by standard ChIP-seq specifically exclude BURs, showing a sharp indentation at the peak of BUR distribution, SATB1-binding profiles by urea ChIP-seq coincide with BUR distribution (*Figure 2D*). Heatmap analyses were also conducted to examine SATB1-binding sites among all BURs and BURs occurrence in all SATB1-bound sites in thymocytes (*Figure 2E*). Consistent with results shown in *Figure 2C and D*, the heatmap study shows that the majority of SATB1-binding sites coincide with BURs. Also, SATB1 binds a small subpopulation of BURs in thymocytes (see below for cell-type-dependent SATB1 occupancy of BURs). These results indicate that SATB1 predominantly and directly tethers BURs genome-wide in vivo to the DNase 1-inaccessible nuclear substructure which is depleted in CTCF and enhancers. In other words, the SATB1-binding profiles obtained by standard and urea ChIP-seq methods are mutually exclusive. We conclude that our urea ChIP-seq, which solely captures DNA with its directly bound proteins, unmasks SATB1 direct binding profiles missed by standard ChIP-seq.

## CTCF-binding profiles remain unchanged between urea and standard ChIP-seq

To investigate the potential cause for the discrepancy in the SATB1-binding profiles between the two ChIP-seq approaches, we applied the urea ChIP-seq method to other factors and asked whether binding profiles differed between urea and standard ChIP-seq. We investigated the DNA-binding profile for CTCF by urea ChIP-seq and found its profile to be very similar to that obtained by standard ChIP-seq from ENCODE, which largely overlapped with DNase 1 HS peaks (*Davis et al., 2018*) (*Figure 3A, tracks 2, 4, 5, and 6*). Similarly, we also found that urea ChIP-seq generated essentially identical DNA-binding profiles for polycomb repressive complex 2 (PRC2) core subunits, SUZ12, JARID2, and EZH2 as generated by others using standard ChIP-seq (*Peng et al., 2009*) (*Figure 3—figure supplement 1*). In line with recent studies (*Feng et al., 2022*; *Zelenka et al., 2022*; *Wang et al., 2023*), 37% of SATB1 peaks from standard ChIP-seq (ENCODE) (average of two replicates, *Supplementary file 1*) intersected with CTCF peaks genome-wide. In contrast, only 0.8% of SATB1

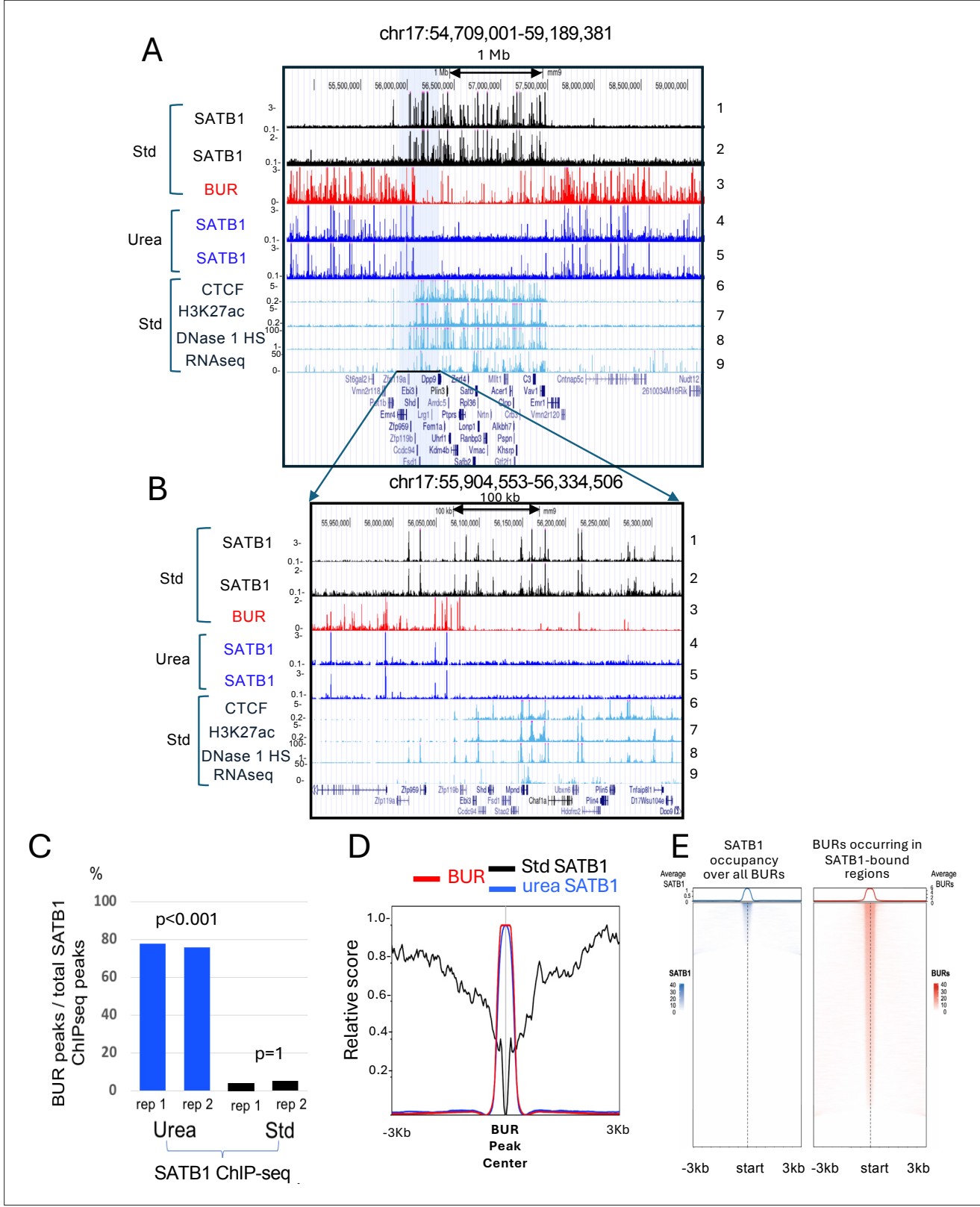

**Figure 2.** Identification of BURs as direct targets of SATB1 in vivo by urea ChIP-seq. (**A**) Contrasting distributions of SATB1-binding profiles determined by urea ChIP-seq (Urea) compared to standard ChIP-seq (Std) in mouse thymocytes. (**B**) A zoomed-in view of the gray-highlighted region shown in (panel A). The track 3 (panels A and B) represents all potential BUR sites serving as a reference for BUR sites. For both panels A and B, two independent urea and standard ChIP-seq experiments show high reproducibility (tracks 1 and 2 and tracks 4 and 5). Track 6 (CTCF) and track 7 (H3K27ac) [ENCODE,

*Figure 2 continued on next page*

*Figure 2 continued*

Bring Ren's Laboratory at the Ludwig Institute for Cancer Research (LICR)] and tracks 8 (DNase 1 HS) and 9 (RNAseq) (ENCODE, the University of Washington (UW) group). (**C**) The percentages of urea (Urea) and standard (Std) SATB1 ChIP-seq peaks that intersect with the BUR reference map (see *Supplementary file 1*). The significance (p values) of the overlap in each of the four cases (urea and standard ChIP-seq samples with two replicates each) with BURs is estimated as the fraction of the overlap counts greater than or equal to the observed count over 1000 random bootstrapped genomic features. p<0.001: none of 1000 random fragments showed greater overlap with BURs than observed in ChIP-seq samples, p=1 means all 1000 random fragments showed greater overlap with BURs than observed in ChIP-seq, indicating avoidance of BURs (see Materials and methods). (**D**) Normalized genome-wide average intensity distribution profiles (normalized relative score) of SATB1 binding sites determined by standard ChIP-seq, urea ChIP-seq as well as BURs, over a ±3 kb window centered at BUR peaks. Relative distance, projection, and Jaccard tests all indicate strong correlation (p<0.05) between urea ChIP-seq peaks for SATB1 and BURs, but no overlap (p<0.05) between standard ChIP-seq peaks for SATB1 and BUR peaks. (**E**) (Left) Heatmaps showing SATB1 ChIP-seq signal intensity centered on all BUR elements. Signal is plotted within ±3 kb of each BUR center using 50 bp bins using EnrichedHeatmap package (v1.30.0). The line plots above the heatmaps display the average SATB1 signal across all regions. Only a subset of BURs exhibits strong SATB1 binding, consistent with selective occupancy. (Right) Heatmaps showing BUR signal intensity centered on SATB1-bound regions (from ChIP-seq peaks). The vast majority of SATB1 binding events are positioned within BURs, as evidenced by sharp, centered BUR signal at nearly all SATB1 peaks. Line plots reflect the average BUR signal profile across all SATB1 sites in each domain class. BUR intensities are derived from in vitro SATB1 BUR-binding assays.

The online version of this article includes the following figure supplement(s) for figure 2:

**Figure supplement 1.** Zoomed-in view of *Figure 2B* showing precise coincidence of SATB1 urea ChIP-seq peaks with a subset of BURs.

peaks from urea ChIP-seq coincided with CTCF peaks (average of two replicas, *Supplementary file 1*), confirming the UCSC browser track views (*Figures 2A and 3A*). These results verify that urea ChIP-seq can accurately identify the DNA-binding sites of chromatin-associated proteins. Thus, urea ChIP-seq, compared to standard ChIP-seq, produces contrasting DNA-binding profiles for SATB1, but this is not necessarily the case for other nuclear proteins. As mentioned in the introduction, it is important to note that these urea ChIP-seq results are supported by a large body of prior binding assay and cytological data for SATB1 that showed SATB1 forms a salt-extraction resistant subnuclear DNA-protein network. Thus, we conclude that for SATB1, urea and standard ChIP-seq produce contrasting SATB1 binding profiles. Importantly, only urea ChIP-seq can faithfully detect direct and specific SATB1 interactions with BURs, suggesting a different mode of chromatin interactions.

## CTCF and SATB1 have distinct and independent binding profiles

Our results above indicate that CTCF and SATB1 *direct* binding sites do not overlap. However, because both CTCF and SATB1 are known to affect chromatin organization, there might be a functional link between the two proteins. Recently, CTCF-binding sites mapped by CUT&Tag were found to be SATB1 independent in specific gene loci (*Wang et al., 2023*). We validated this result by generating the CTCF-binding profiles by urea ChIP-seq in *Satb1^-/-* (SATB1-KO) and *Satb1^+/+* (SATB1-WT) thymocytes. These CTCF-binding profiles showed no major changes depending on SATB1 (*Figure 3A*, *compare tracks 2 and 3*), as shown in a randomly chosen region in chromosome 17 in wild-type (WT) and SATB1-KO thymocytes. This was confirmed by genome-wide analyses which showed that CTCF urea ChIP-seq peak distributions from SATB1-WT and SATB1-KO thymocytes coincided (*Figure 3B*). This indicates that CTCF binding to genomic DNA is SATB1-independent genome-wide.

To corroborate the independence of CTCF/cohesin and SATB1, we next examined whether SATB1 interacts with CTCF and/or cohesin or both, components known to be important for loop formation. Distinct from protein-protein co-immunoprecipitation (co-IP) using whole cell or nuclear extracts, we examined the direct co-binding status on chromatin in vivo of SATB1 and CTCF or cohesin by urea ChIP-Western. Urea-purified crosslinked-chromatin fragments immunoprecipitated with anti-SATB1 antibody (SATB1-bound urea-ChIP fragments) contained SATB1 as expected, detected by western blot of the de-crosslinked urea-ChIP sample. However, neither CTCF nor RAD21 (a cohesin complex component) was detected in the SATB1 urea-ChIP samples. Reciprocal experiments using either CTCF or RAD21-bound urea ChIP samples confirmed lack of SATB1 co-binding to these chromatin fragments (*Figure 3C*, *Figure 3—figure supplement 2A*, *Figure 3—source data 1*, *Figure 3—source data 2*). We confirmed that a similar amount of urea-purified crosslinking chromatin was used for each ChIP-Western sample with an anti-H3 antibody (which is, as expected, retained on urea-purified crosslinked chromatin). Therefore, these results from urea ChIP-Western indicate that at least for the direct DNA co-binding status of SATB1 with either CTCF or RAD21, SATB1 does not co-bind chromatin with

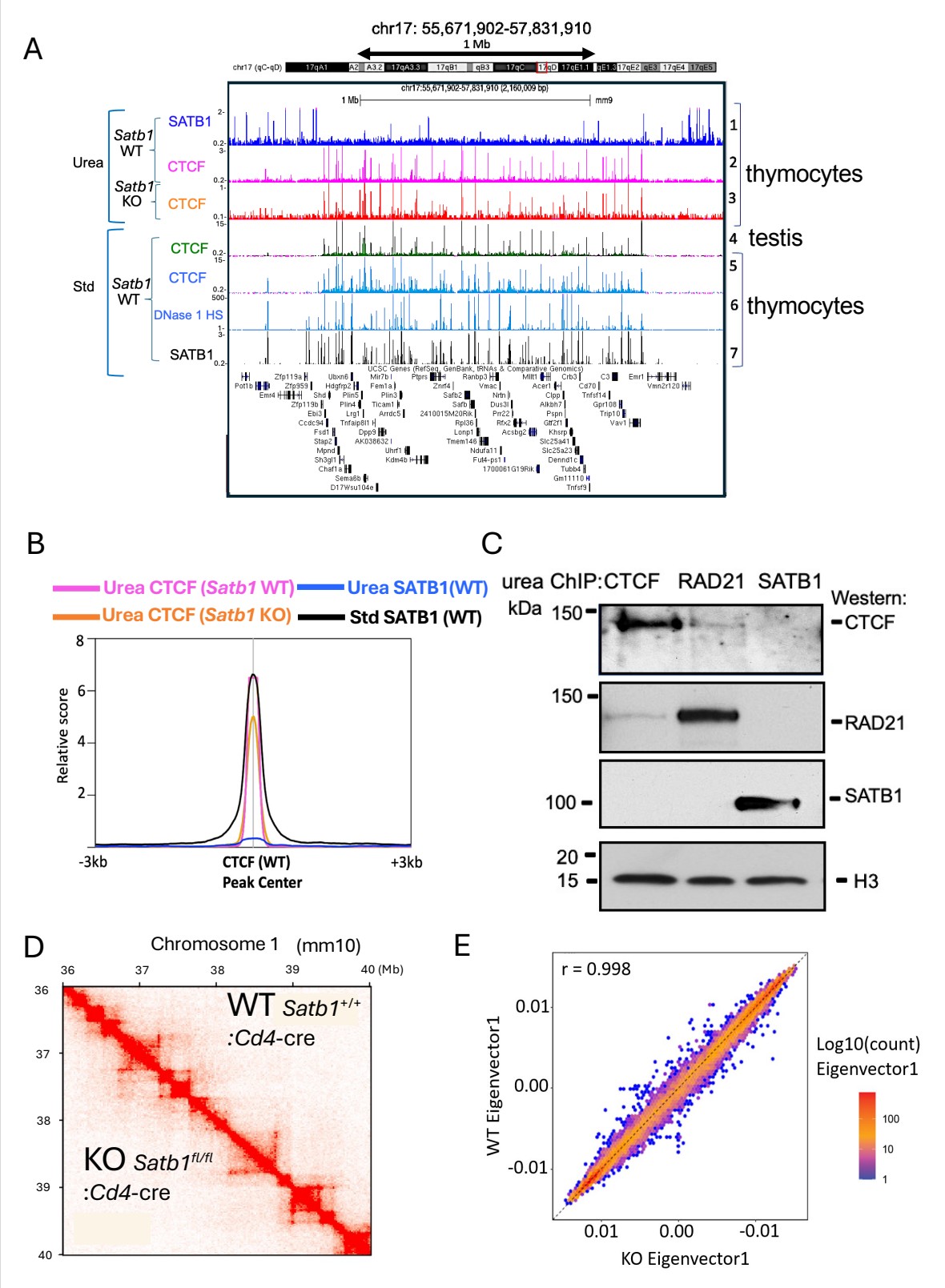

**Figure 3.** Identical CTCF-binding profiles are produced by urea and standard ChIP-seq, independent of SATB1. (**A**) Urea ChIP-seq (Urea) profiles for CTCF in *Satb1*^(+/+) (WT) thymocytes and *Satb1*^(-/-) (KO) thymocytes (track 2 and 3), and for SATB1 in WT thymocytes (track 1). These binding patterns were compared against CTCF-binding profiles generated by standard ChIP-seq (Std) from ENCODE (tracks 4 and 5) for testis and thymocytes (ENCODE, LICR), respectively, along with the DNase1 HS profile from ENCODE, UW (track 6) as well as the SATB1-binding profile from standard ChIP-seq (track

*Figure 3 continued on next page*

*Figure 3 continued*

7). (**B**) Genome-wide average intensity distribution (relative score) of peaks from CTCF and SATB1 urea ChIP-seq for *Satb1*<sup>+/+</sup> thymocytes (WT), CTCF urea ChIP-seq for *Satb1*<sup>-/-</sup> thymocytes (KO), and SATB1 standard ChIP-seq (WT) over a ±3 kb window centered at CTCF peaks (WT). Relative distance, projection, and Jaccard tests all indicate strong correlation (p<0.05) between CTCF urea ChIP-seq peaks for WT and KO thymocytes, between these peaks and standard ChIP-seq peaks for SATB1, but no overlap (p<0.05) between SATB1 urea ChIP-seq peaks. (**C**) Urea-ChIP-Western results for CTCF, RAD21, and SATB1, showing that SATB1 does not associate with CTCF nor RAD21 on chromatin. Also, CTCF and RAD21 co-binding to chromatin appears to be infrequent, suggested by the weak ChIP-Western signals. (**D**) Hi-C interaction heatmaps for *Satb1*<sup>+/+</sup>:*Cd4*-Cre (WT) (top right triangle) and *Satb1*<sup>fl/fl</sup>:*Cd4*-Cre (KO) thymocytes (bottom left triangle) show essentially identical patterns. (**E**) Genome-wide comparison of distributions of cis-Eigenvector 1 values shows no difference between WT and KO thymocytes (high correlation at $r=0.998$), indicating that genomic compartmentalization is unaffected in *Satb1* KO thymocytes. Experiments were done in two replicates each for WT and KO samples. The data shown in D and E were generated by combining data from the replicates for each condition.

The online version of this article includes the following source data and figure supplement(s) for figure 3:

**Source data 1.** Original raw data for ChIP-Western results shown in *Figure 3C* (labelled).

**Source data 2.** Original raw data for ChIP-Western results shown in *Figure 3C*.

**Figure supplement 1.** Similar DNA binding profiles for PRC2 core subunits obtained by standard ChIP-seq and Urea ChIP-seq.

**Figure supplement 2.** SATB1 does not directly co-bind chromatin with CTCF and RAD21 or has a role in TAD formation.

either of these proteins (*Figure 3C*). These results are consistent with mutually exclusive direct binding sites of CTCF and SATB1 (*Figure 3A and B*).

If cohesin and CTCF bind directly to chromatin for loop formation according to the loop extrusion model, then we expect that the urea ChIP-Western results will show that cohesin-bound DNA fragments are co-bound by CTCF at the base of loops. As expected, we do find co-binding; however, fragments co-bound by RAD21 and CTCF represent only a small fraction of either total RAD21-bound or CTCF-bound chromatin fragments (*Figure 3C*). These results show that CTCF and RAD21 direct co-binding to chromatin occurs much less frequently than their individual binding to chromatin. This result agrees with a recent report which used super-resolution imaging of individual cells to show that the stable CTCF-mediated loops are rare, and >95% of the time, the loops are either only partially extruded, or no loops are formed (*Gabriele et al., 2022*).

Given that CTCF deletion causes most TADs to disappear, we next examined whether SATB1 depletion has any effects on TAD formation. We compared bulk Hi-C results between thymocytes isolated from wild-type (*Satb1*<sup>+/+</sup>:*Cd4-Cre*) and SATB1-deficient (*Satb1*<sup>fl/fl</sup>:*Cd4-Cre*) mice in duplicates (*Figure 3—figure supplement 2B*). Our Hi-C results revealed largely indistinguishable TAD profiles between the two thymocytes (*Figure 3D*) examined at 10 kb resolution. Neither TAD number nor TAD sizes were changed by SATB1 depletion from thymocytes (*Figure 3—figure supplement 2C*). Intra-TAD interactivity (the sum of Hi-C chromatin contacts inside the TAD) also remained mostly unchanged (*Figure 3—figure supplement 2D*). These bulk Hi-C results upon SATB1 deletion are in good agreement with recently published studies (*Feng et al., 2022*; *Zelenka et al., 2022*). Similar to the results from the CTCF depletion study (*Nora et al., 2017*), SATB1 depletion also did not alter segregation of active (A) and inactive chromatin compartmentalization (B) (*Figure 3E* and *Figure 3—figure supplement 2E*). Furthermore, our scaling plot displaying how the Hi-C contacts decrease as a function of genomic distance shows that SATB1 depletion does not affect chromosome compaction (*Figure 3—figure supplement 2F*). These Hi-C re,sults taken together with SATB1-independent CTCF-binding to DNA and non-overlapping direct genomic DNA-binding of SATB1 and CTCF/cohesin, strongly suggest that SATB1 and CTCF are two independent proteins that have distinct roles in chromatin organization, consistent with SATB1 having no roles in TAD organization.

## Non-specific peaks appear in *Satb1*<sup>-/-</sup> thymocytes by standard SATB1 ChIP-seq

An important question is whether SATB1-binding profiles obtained by standard ChIP-seq are strictly SATB1 dependent. If so, these binding sites likely reflect real but indirect SATB1 association with chromatin. To address this question, we performed standard ChIP-seq on *Satb1*<sup>-/-</sup> (KO) thymocytes using two different anti-SATB1 antibodies (1583 and ab109112, both of which are validated for immunoprecipitation). The results are highlighted in two randomly chosen ~3 Mb regions in chromosome 17 and in the X chromosome, each containing a gene-rich region surrounded by gene-poor regions.

Both antibodies generated similar peaks in KO thymocytes (KO peaks) by standard ChIP-seq. These KO peaks (false positives) were predominantly detected in gene-rich, open chromatin regions in KO thymocytes, and many of them coincided with peaks originally detected in WT thymocytes (WT peaks; *Figure 4—figure supplement 1A and B, top and bottom, tracks 9 and 10*). This is in agreement with an earlier report that ChIP-seq generates false-positive peaks in highly accessible, transcriptionally active chromatin regions (*Jain et al., 2015*; *Park et al., 2013*; *Teytelman et al., 2013*; *Meyer and Liu, 2014*; *Baumgarten and Bryant, 2022*). Intersection analyses with deepTools show that 4.1% of SATB1 standard ChIP-seq peaks obtained from our experiments are false-positive peaks (based on average of two standard KO ChIP-seq samples; *Supplementary files 3 and 4*). Besides those KO peaks that overlapped with WT peaks, there were several thousand KO-specific peaks generated by standard ChIP-seq in our study. In contrast, SATB1 urea ChIP-seq had no false-positive peaks (0.0%). Also, only ~100 KO-specific peaks (not overlapping with WT peaks) were detected (average of two samples) compared to ~40,000 total WT peaks from SATB1 urea ChIP-seq (*Supplementary files 3 and 4*). Therefore, validation using knockout cells is critical to distinguish real peaks that depend on SATB1 from SATB1-independent phantom peaks derived from the standard ChIP-seq method. The percentages of phantom peaks in standard ChIP-seq likely depend on the specificity of antibodies used and the condition of sample preparation.

After validation with SATB1-KO cells, we conclude that most SATB1 sites detected from standard ChIP-seq are bound by SATB1, but most likely through indirect binding, as these peaks are not detected by urea ChIP-seq, which removes any indirect binding proteins by urea ultracentrifugation. SATB1 proteins are found in high salt-resistant fraction as well as salt-extracted fraction (*de Belle et al., 1998*). Thus, it is possible that soluble SATB1 may associate with 'open' chromatin. In order to further understand these differences and potential proximity of SATB1 binding sites detected by the two ChIP-seq methods, we focused on three individual gene loci, that is *Rag1* and *Rag2* genes (*Hao et al., 2015*), *Foxp3* (*Kitagawa et al., 2017*), and *Runx3* (*Kakugawa et al., 2017*), for which specific SATB1-binding sites by standard ChIP-seq have been reported (*Figure 4*). In addition, these genes are known to be regulated by SATB1 in thymocytes and are thus excellent candidate regions to investigate how SATB1 is regulating chromatin and gene expression. By either direct urea-ChIP or standard ChIP, we found SATB1-bound sites within all three loci (*Rag1/2*, *Foxp3*, and *Runx3*) to be SATB1-dependent. These sites were detected only in WT (*Figure 4, track 3*) but not in SATB1 KO-thymocytes (*Figure 4, tracks 7 and 9 for Rag/Rag2 and Runx3, and tracks 9 and 11 for Foxp3*). In these regions, input sample (without antibodies) had no peaks (*Figure 4, tracks 8 and 10 for Rag1/Rag2 and Runx3, and tracks 10 and 12 for Foxp3*). For the *Rag1* and *Rag2* loci, SATB1 has been shown by standard ChIP-seq to bind the anti-silencer element (ASE) and this binding is thought to counteract the activity of an intergenic silencer, thus promoting expression of *Rag 1* and *Rag 2* (*Hao et al., 2015*). During the development of Foxp3[+]-regulatory T cells (Treg) in the thymus, SATB1 expression in Treg precursor cells is essential for the future establishment of a Treg cell–specific super-enhancers at the *Foxp3* locus in Treg cells (*Kitagawa et al., 2017*). Thus, thymocyte precursor-specific SATB1 deficiency, using *Satb1*[fl/fl]*:Cd4-Cre* mice but not *Satb1*[fl/fl]*:Foxp3*-Cre mice, impairs Treg super-enhancer activation and expression of *Foxp3* (*Kitagawa et al., 2017*). Impaired Treg cell development results in autoimmune disease (*Kakugawa et al., 2017*; *Kitagawa et al., 2017*; *Tanaka et al., 2017*; *Kondo et al., 2016*). Finally, SATB1 is also essential for the generation of CD4[+] and CD8[+] T cells and NKT cells (*Kakugawa et al., 2017*). In response to T cell receptor signaling by MHC class I and II, SATB1 regulates enhancer function in gene loci encoding lineage-specifying factors (e.g. *Runx3*) and directs lineage-specific transcriptional programs in the thymus (*Kakugawa et al., 2017*). Strikingly, in these three regions, SATB1-bound BURs, as detected by urea ChIP-seq, are found infrequently and appear as isolated single peaks within 30–50 kb regions. Such SATB1-bound BUR sites are located close to but not overlapping with the cluster of enhancers marked by H3K27ac and H3Kme1 modifications. These SATB1-bound BUR sites reside within 3 kb of the nearest putative SATB1-bound enhancers (detected by standard ChIP-seq) for the *Rag1/Rag2* and *Runx3* loci as well as from the super-enhancer that becomes established at the *Foxp3* loci in matured Treg cells. Results from these three loci show that, while the SATB1-bound peaks generated by urea ChIP and standard ChIP-seq did not overlap, they were nonetheless proximal to each other, and both were SATB1 dependent. The two distinct ChIP-seq protocols are likely detecting two different modes of DNA binding for SATB1 that is direct binding to BURs and indirect binding to open chromatin regions.

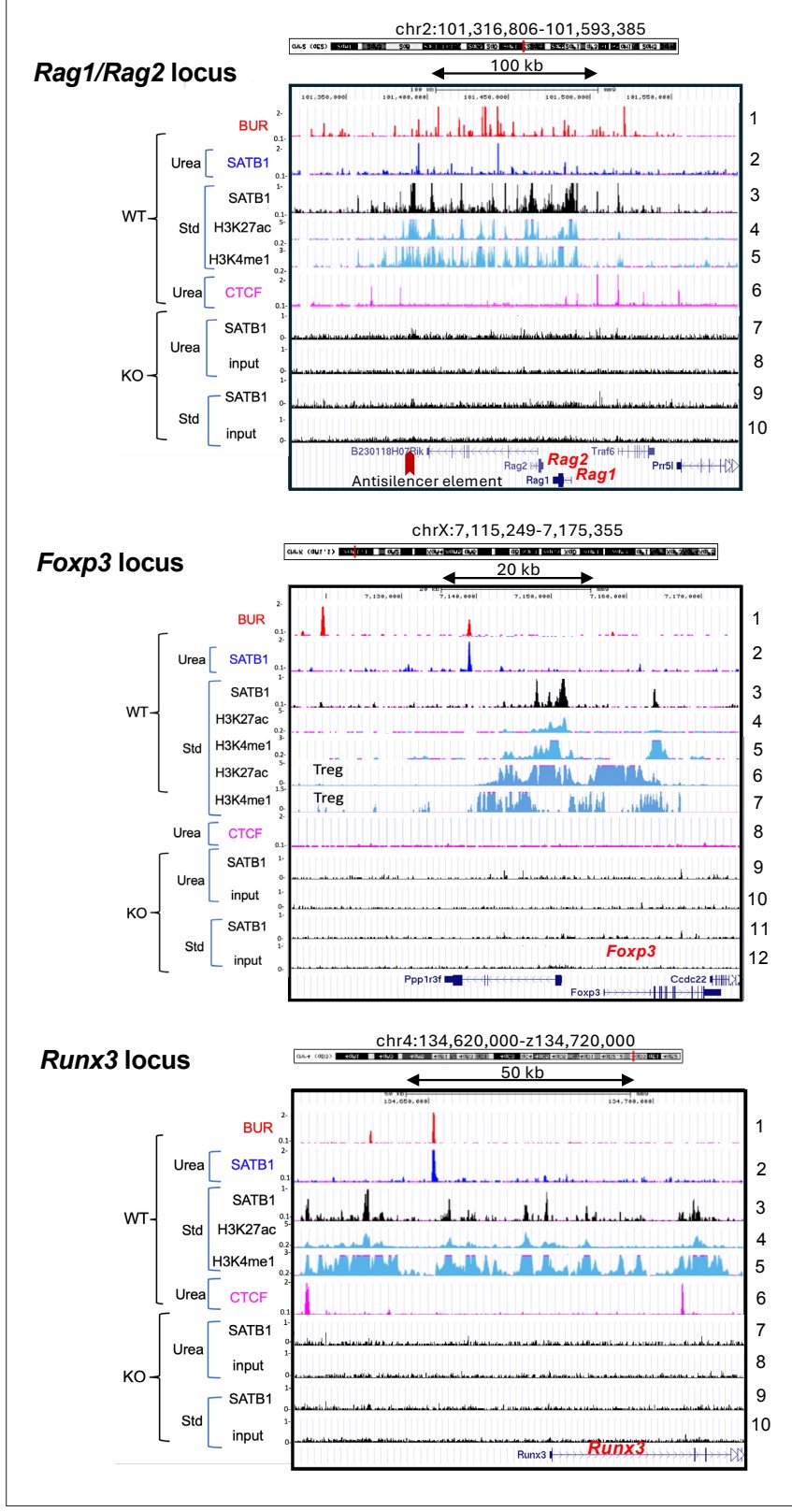

**Figure 4.** SATB1-bound sites identified by standard ChIP-seq at regulatory regions are SATB1 dependent. SATB1-binding profiles in immature thymocytes determined by urea ChIP-seq (Urea) and standard ChIP-seq (Std) covering three loci at *Rag1/Rag2* (top) *Foxp3* (middle), and *Runx3* (bottom) are shown. For each of the three loci, the standard SATB1 ChIP-seq generates peaks in *Satb1⁺/⁺* (WT) thymocytes (track 3), but not in *Satb1⁻/⁻* (KO) thymocytes

*Figure 4 continued on next page*

*Figure 4 continued*

(tracks 9 for *Rag 1/2* and *Runx3* and tracks 11 for *Foxp3*). Input chromatin for standard ChIP-seq in these regions did not produce any major peaks (track 10 in *Rag1/Rag2* and *Runx3* and track 12 in *Foxp3*). The SATB1 peaks from standard ChIP-seq in *Rag1/Rag2* and *Runx3* loci largely overlap with enhancers (marked by H3K4me1 and H3K27ac). At the *Foxp3* locus in immature thymocytes, SATB1 peaks (track 3) appear in a region marked by low levels of H3K27ac and H3Kme1 (tracks 4 and 5) where super-enhancer is established upon differentiation of thymocytes into Treg cells (tracks 6 and 7; *Kitagawa et al., 2017*). SATB1-bound peaks produced by urea ChIP-seq (track 2) are consistently located in distinct positions from those peaks obtained by standard ChIP-seq in these gene loci. Urea ChIP-seq did not generate peaks in KO thymocytes with anti-SATB1 antibody, and input chromatin also did not produce any peaks (tracks 7 and 8 for *Rag1/Rag2* and *Runx3*, and track 9 and 10 for *Foxp3*). H3K27ac and H3Kme1 data (tracks 4 and 5) are from Encode, generated by Bing Ren's laboratory at LICR, and data for Treg shown in (tracks 6 and 7) are from Kitagawa (*Kitagawa et al., 2017*).

The online version of this article includes the following figure supplement(s) for figure 4:

**Figure supplement 1.** Standard ChIP-seq for SATB1 produces non-specific peaks in Satb1-/- thymocytes.

## Long-distance chromatin interactions involving a BUR depend on SATB1

We next explored whether there is any three-dimensional relationship between SATB1-bound BURs detected by urea-ChIP-seq and SATB1-bound sites in the open chromatin region detected by standard ChIP-seq. We hypothesized that SATB1-rich nuclear substructure where BURs are directly bound to SATB1 may interact with open chromatin regions by SATB1's indirect association with their chromatin-bound protein complexes. If so, such interactions would necessarily be dependent on SATB1, and they might be correlated to gene expression. If so, such interactions would necessarily be dependent on SATB1, and they might be correlated to gene expression. To address these questions, we studied a gene-rich region in chromosome 2 that contains the *Rag1* and *Rag2* genes whose expression is SATB1 dependent (*Hao et al., 2015*). We first examined genome-wide interactions from a SATB1-bound BUR that appeared as a strong peak as the viewpoint or bait (BUR-1) (*Figure 4, top track 2 urea ChIP-seq as well as Figure 5A and Figure 5B*). The BUR-1 primer sequence is located near the border between the 3.9 Mb gene-desert region and the 5.8 Mb gene-rich region on chromosome 2 where the major BUR bound to SATB1 is located. BUR-1 resides near *Rag1* and *Rag2* genes in the 5.8 Mb gene-rich region and 1.4 kb away from the antisilencer element (ASE) (*Figure 4, top, track2*), which is essential for proper *Rag1* and *Rag2* expression in double-positive thymocytes and for differentiation of single-positive thymocytes (*Yannoutsos et al., 2004*). It is important to note that BUR-1 (as well as other SATB1-bound BURs in the 5.8 Mb gene-rich region) precisely avoids DNase 1 HS peaks, indicating that it resides in discrete DNase 1-inaccessible chromatin in otherwise active and accessible regions (see the Zoom-in view in *Figure 5B*).

To study interactions from the SATB1-bound BUR, we performed 4C-seq using urea-purified chromatin (urea 4C-seq, one-to-all) to better access and capture BUR-mediated interactions, which are expected to emanate from the SATB1 subnuclear architectural structure. In our urea 4C-seq, we used *Hind*III for initial digestion. The 6-base cutter, *Hind*III, was used instead of a 4-base cutter like *Dpn*II (frequently used for 4C-seq), because multiple *Dpn*II sites are often found within individual SATB1-bound BURs, thus producing small fragments, likely disrupting SATB1-binding to BURs. The 'BUR-1' bait region at a SATB1-bound BUR is designed at the end of the *Hind*III fragment, which is adjacent to the *Hind*III fragment containing ASE. We examined interaction events from the fragment containing BUR-1 and its interaction sites were mapped on the genome for visualization.

While urea purification removes indirectly associated protein and thus may reduce some chromatin interactions, we speculated that the overall 3D chromatin structure of fixed chromatin will still retain its proximity events, especially in the environment of chromatin which is strongly anchored to the nuclear substructure (as we predicted for SATB1 scaffold anchoring at BURs) to enable detection in urea 4C-seq. In the ChIP-seq protocol (in general), we expect that in the absence of a ligation step as is routinely done in 4 C protocols (as described below), subsequent steps for ChIP-seq (sonication, stringent washes of immunoprecipitated fragments, etc) result in dissociation of interacting fragments. We carried out urea 4C-seq experiments to test if we could indeed detect such strong SATB1-mediated 3D interactions. Briefly, for urea 4C-seq (outlined in *Figure 5—figure supplement 1*), crosslinked chromatin purified by urea ultracentrifugation is first digested with a restriction enzyme (*Hind*III). Digestion is followed by ligation of chromatin fragments whose physical proximity status in

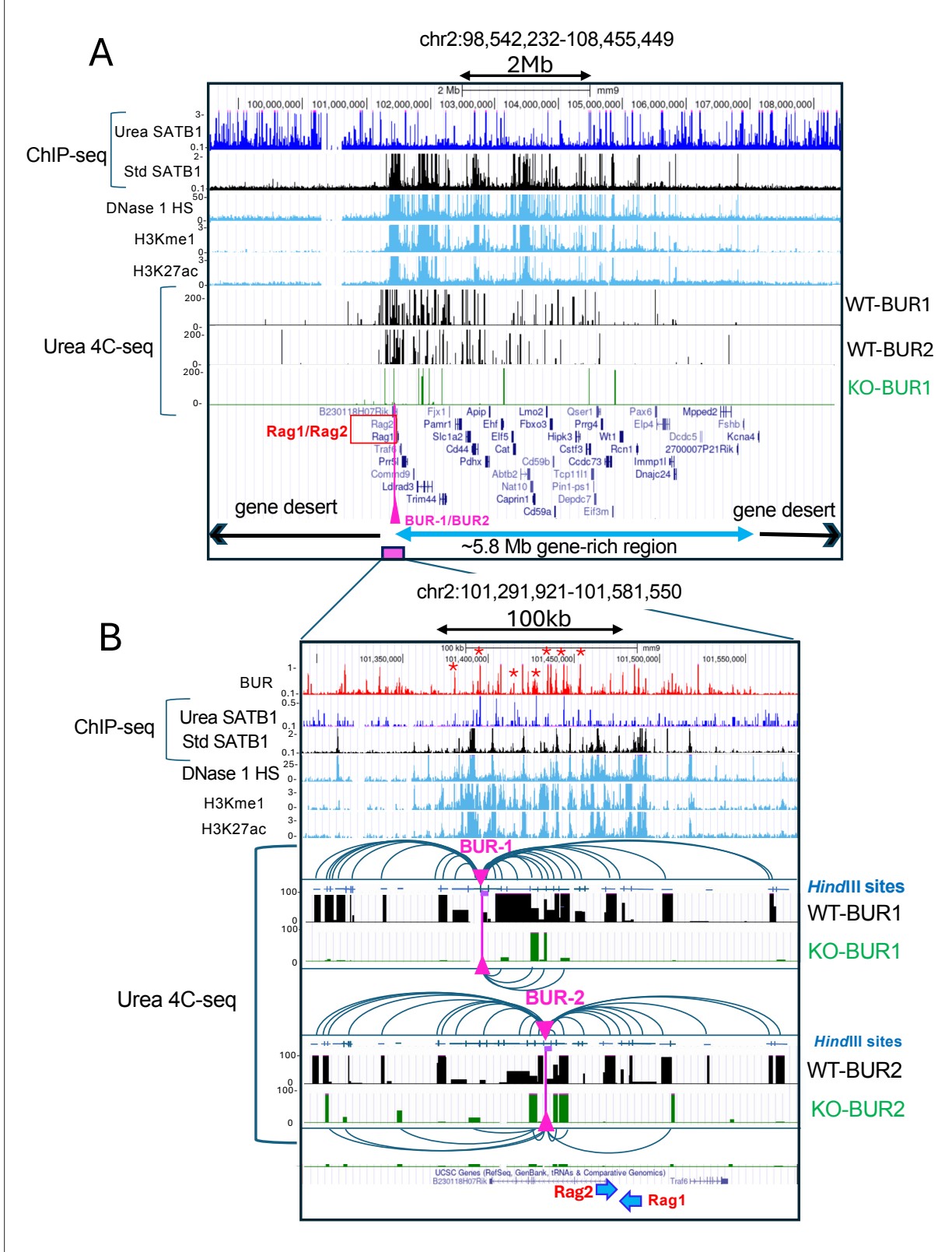

**Figure 5.** SATB1 mediates dense chromatin interactions over *Rag1/2*-containing gene-rich region. (**A**) Chromatin interactions detected by urea 4C-seq using a SATB1-bound BUR (BUR-1) and another site (BUR-2) as baits. These baits are located near the border of a gene-poor region and a gene-rich region containing SATB1-regulated *Rag1* and *Rag2* genes (pink vertical bar with arrowhead). BUR-2 was found to be one of the interacting sites of BUR-1. Therefore, it was used as a reciprocal control for BUR-1 to check the reproducibility of BUR-1 interactions. We used *Satb1*$^{+/+}$ (WT) and *Satb1*$^{-/-}$ (KO)

*Figure 5 continued on next page*

*Figure 5 continued*

thymocytes from 2 weeks old mice for urea 4C-seq. BUR-1 and BUR-2 interact with many sites within the ~5.8 Mb gene-rich region, containing *Rag1* and *Rag2* (tracks WT-BUR1 and WT-BUR2). These interactions were greatly reduced upon SATB1 deletion (track KO-BUR1). (**B**) A zoom-in view, focusing on the *Rag1* and *Rag2* region, shows that BUR-1 interacts extensively over this region in WT thymocytes (shown by parabolas). Such interactions were virtually diminished in the absence of SATB1 (KO-BUR1 and KO-BUR2 tracks). The pink vertical bars with arrowheads indicate the positions of the BUR-1 and BUR-2 bait primers. Each of these primer sequences is located at the end of the *Hind*III fragment containing the primer. These *Hind*III fragments are marked by thick purple bars adjacent to pink arrowheads pointing to primer positions. The map of *Hind*III fragments is shown for interacting regions (track *Hind*III sites). BUR-1 and BUR-2, along with other BURs bound by SATB1 (as confirmed by SATB1 urea ChIP-seq peaks and the BUR reference) that interact with BUR-1 or BUR-2 are marked by small red stars in the BUR track. Note: Reads to the *Hind*III fragment immediately adjacent to each of the bait primers (BUR-1 and BURs) were removed as they comprised more than 2% of the total reads, typically derived from re-ligation of digested chromatin fragments to their original neighboring sequences and/or from a small undigested chromatin fraction.

The online version of this article includes the following figure supplement(s) for figure 5:

**Figure supplement 1.** Urea 4C-seq protocol.

**Figure supplement 2.** Long-range chromatin interactions from BUR-1 traverse TADs.

**Figure supplement 3.** Long-distance interactions covering distal gene-rich regions by BUR-1.

**Figure supplement 4.** SATB1-dependent genes identified within the three gene-rich regions that interact with BUR-1.

3D space was fixed at the time of formaldehyde crosslinking. After these steps, DNA was purified. Beyond this point, urea-4C-seq differs from the commonly used 4C-seq called circular chromosome conformation capture (referred to here as circular 4C-seq) (*Ben Zouari et al., 2019*; *de Wit et al., 2013*; *Geeven et al., 2018*; *Krijger et al., 2020*; *Noordermeer et al., 2011*; *Wijchers et al., 2016*). In circular 4C-seq, ligated DNA fragments are circularized after second restriction enzyme digestion before PCR amplification. In contrast, in urea 4C-seq, a single-strand DNA capture strategy was used to isolate a specific pool of single-strand DNA, each containing a ligation-captured DNA fragment surrounded by the bait fragment and the DNA linker (see details in *Figure 5—figure supplement 1* legend and Materials and methods). The captured DNA sequences in the single-strand DNA were PCR amplified using a primer from the bait sequence and another primer from the DNA linker, thus avoiding amplification of any non-specific DNA fragments. This was confirmed by the lack of PCR products using either primer alone (data not shown). This critical step results in highly specific signals with very low background signals. The high signal-to-noise ratio in urea 4C-seq contrasts with uniform and strong genome-wide background signals typically seen with circular 4C-seq, which are thought to reflect random chromatin interaction events. Some non-interacting sequences could also be trapped at the circularization step, and even if such incidence is rare, would result in background signals upon PCR amplification. Because the background signals for urea 4C-seq are extremely low in contrast to circular 4C-seq, the visual appearances of the interacting profiles from the two methods are greatly different. Nevertheless, urea 4C-seq detects large-scale interactions involving BURs that are strictly dependent on SATB1 as described below.

In wild-type thymocytes, we detected BUR-1 making extensive long-distance interactions covering most of the ~5.8 Mb adjacent gene-rich region (*Figure 5A*). Interestingly, BUR-1 interactions are found to be scarce over the gene-desert region upstream of BUR-1, despite the presence of many SATB1-bound BURs in this region (*Figure 5A, top track for urea SATB1 ChIP-seq*, *Figure 5—figure supplement 2*). Though 4C-seq can detect both up- and downstream interactions from a chosen single bait site, we observed heavily direction-biased interactions of BUR-1 with the gene-rich region, perhaps indicative of its involvement in gene regulation. BUR-1 interactions were found particularly concentrated over the ~3 Mb gene-rich region, containing SATB1-dependent *Rag1* and *Rag2* genes. These dense and strong interaction signals must not be due to their spatial proximity from the bait because similar dense interactions were not detected with the SATB1-KO cells. In fact, most of the BUR-1 interactions were absent in SATB1-KO cells (*Figure 5A, track KO-BUR1*), indicating that SATB1 is required for the numerous long-distance interactions over the gene-rich region. We also validated these interactions using a second bait (BUR-2) as an experimental control as BUR-1 interacts with the *Hind*III fragment containing BUR-2, located in the second intron of *B230118H07Rik* gene (see *Figure 5B* for specific location). This fragment contains two strong BUR peaks. Although weakly, SATB1 binds to one of these BURs. Hence, urea 4C-seq using BUR-2 serves as a reciprocal experiment of BUR-1 to validate reproducibility of the BUR-1 interaction profile. By comparing with the Hi-C heatmap in this 5.8 Mb region, we found that BUR-1 interactions traverse multiple TADs and are not constrained by any

specific TADs (*Figure 5—figure supplement 2*), further supporting the CTCF-independent nature of these SATB1-mediated interactions. Within this gene-rich region, numerous interactions from SATB1-bound BUR-1 were detected within DNase 1 HS regions containing enhancers marked by H3K27ac and H3Kme1 modifications. Therefore, dense chromatin looping events involving these SATB1-bound BURs indicate a highly interactive chromatin architecture for the gene-rich region, bringing together many gene regulatory sequences into proximity. We further examined a wider region covering >44 Mb that includes distal neighboring gene-rich regions (*Figure 5* and *Figure 5—figure supplement 3A and B*). Although interaction frequencies were lower than that within the first ~5.8 Mb gene-rich region, we detected BUR-1 interactions with further distal gene-rich region coinciding with enhancer-rich DNase 1 HS regions, where clusters of SATB1 binding sites were detected by standard ChIP-seq, skipping over gene-poor regions and a gene-rich region containing transcriptionally silent olfactory receptor genes.

A zoomed-in view (chr2:101,308,846–101,599,827; 291 Kb) of BUR-1 interactions confirmed a marked difference in BUR-1 interactions (~200 Kb) over the *Rag1* and *Rag 2* loci between wild-type and *Satb1⁻ᐟ⁻* thymocytes, confirming BUR-1 interactions require SATB1 (*Figure 5B*). In addition to many BUR-1 interacting chromatin fragments close to or containing enhancers (marked by H3K27ac and H3Kme1), some of these fragments contain additional BUR sites, including BUR-2 adjacent to enhancer(s) (*Figure 5B*, peaks labeled with red stars). BUR-2 interacted heavily over the *Rag1* and *Rag2* loci in wild-type thymocytes and well reproduced the interactions observed with BUR-1, consistent with the larger region (*Figure 5A*), suggesting that both BUR-1 and BUR-2 are part of the common interaction network that requires SATB1. Concerning the correlation of chromatin interactions and gene expression, among 49 genes identified with official gene symbols mapped in the 5.8 Mb gene rich region, 3 genes display strong SATB1-dependent expression (adjusted p value<0.05) from RNA-seq data (*Zelenka et al., 2022*). *Rag1* and *Rag2* were strongly downregulated and *Prrg4* was upregulated upon SATB1 depletion. In the two neighboring gene-rich regions that are also engaged in chromatin interactions with BUR-1 (*Figure 5—figure supplement 3A and B*), expression of 10 additional genes with important roles in thymocytes and T cells were found strongly SATB1 dependent. These genes were either up or downregulated upon SATB1 deletion in thymocytes based on RNA-seq data (*Figure 5—figure supplement 4*). Collectively, these data suggest a strong correlation between SATB1-dependent gene expression and extensive looping events over long distances linking SATB1-bound BURs and regulatory regions in the open chromatin regions. Therefore, SATB1 function on chromatin is not limited to merely connecting a specific target gene locus with nearby enhancers and promoters within TADs. Instead, SATB1 appears to form a large interaction network covering the entire gene-rich regions.

## SATB1 binds BURs in a cell-type-dependent manner

We next examined whether SATB1 binds different subsets of BURs depending on cell type, by studying SATB1-binding profiles in three different types of mouse tissues – thymus, frontal cortex, and skin – that contain SATB1-expressing cells. SATB1 expression is primarily restricted to adult progenitor cells, such as thymocytes and in the basal layer of epidermis. SATB1 is also expressed in differentiated neurons in specific brain subregions (e.g. frontal cortex and amygdala). We compared UCSC browser images of two selected regions, a ~2.4 Mb region in chromosome 6 containing the *Cd4* locus expressed in thymocytes (*Figure 6A*) and a 406 kb region in chromosome 5 containing *Gabra2* and *Gabrg1* loci expressed in the mammalian brain (*Figure 6B*). BURs were most frequently bound by SATB1 in brain compared to thymocytes, and the least frequently bound in skin. Whereas SATB1-bound BURs show cell-type dependency, CTCF binding sites that largely coincide with DNase1 HS were largely similar between these three types of cells (*Figure 6A*, *tracks 11-13*).

We analyzed whether BURs are targeted by SATB1 depending on cell type. Consistent with the cell-type-specific function of SATB1, we observed cell-type-dependent differences in BUR occupancy by SATB1. For all three types of cells, high percentages of urea ChIP-seq peaks (close to 80%, *Supplementary file 1*) correspond to BURs, as revealed by peak intersection studies using deepTool (*Ramírez et al., 2016*) and comparing to the BUR reference map (*Figure 6C* and *Supplementary file 1*). Despite similar total aligned sequence reads in urea ChIP-seq, the number of SATB1-bound BUR peaks greatly differed between cell types (*Supplementary file 2*). Among 238,380 mapped BURs (q<0.01), the percentage of SATB1-bound BURs in skin, thymocytes, and brain was 3.7%, 13.5%, and

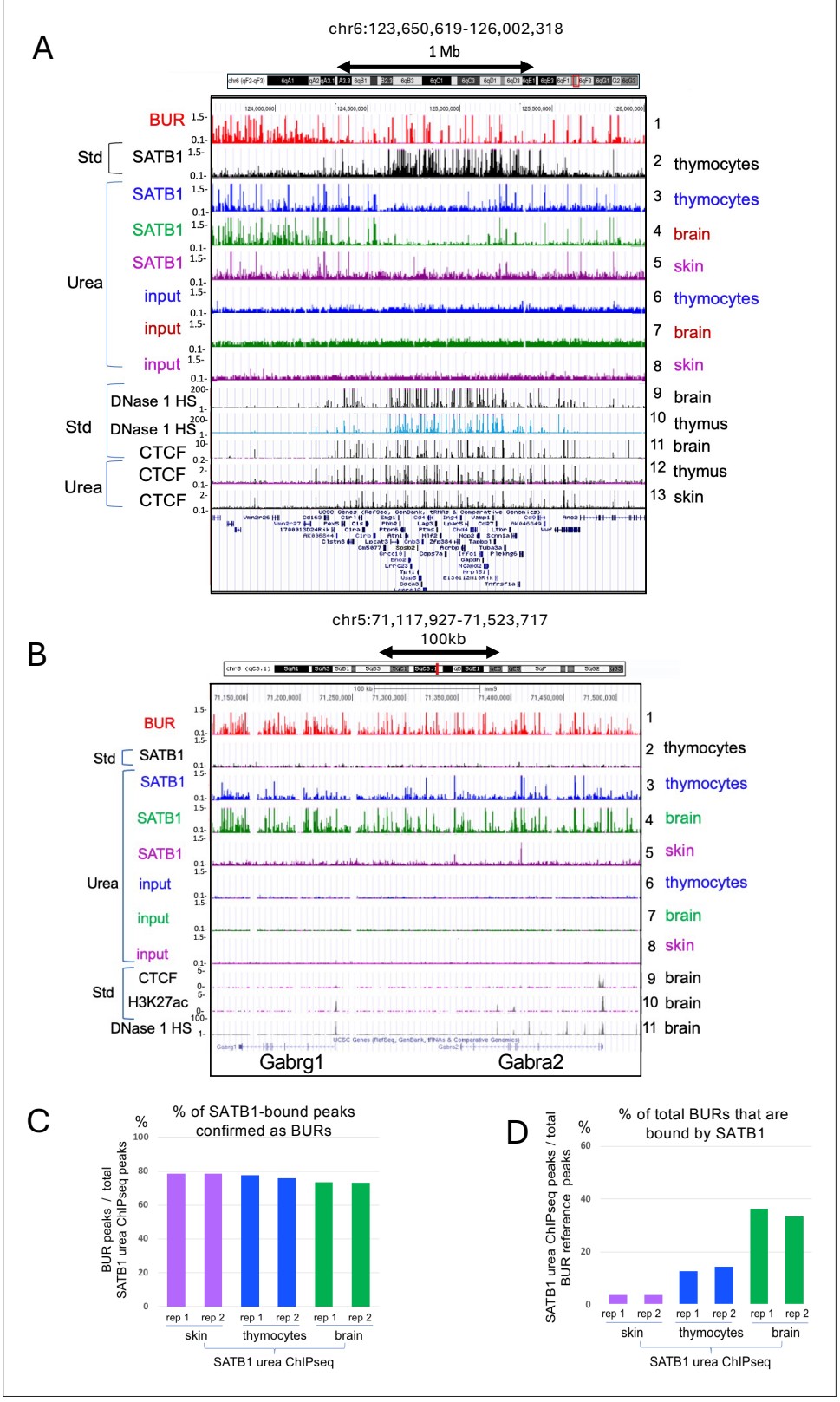

**Figure 6.** SATB1 binds to BURs in a cell-type-dependent manner. (**A**) SATB1-binding profiles in a ~2.4 Mb region in chromosome 6 containing *Cd4* in a gene-rich region from standard ChIP-seq (Std) (track 2) and urea ChIP-seq (Urea) (tracks 3–8). Urea ChIP-seq-derived SATB1 profiles in thymocytes (track 3), brain (track 4), and skin (track 5) show differences in peak numbers depending on cell type (see *Supplementary files 1 and 2*). This contrasts with

*Figure 6 continued on next page*

*Figure 6 continued*

mostly identical CTCF patterns in these cell types (tracks 11–13). (**B**) SATB1-binding profiles in a 406 kb region in chromosome 5 containing *Gabra2* and *Gabrg1*, highly expressed in brain, also show much increased BUR binding in brain, compared to thymocytes and skin similar to A. In this region, there is no SATB1-binding site detected by standard ChIP-seq (track 2), and CTCF binding sites are mostly absent (track 9). (**C**) Percentages of SATB1 urea-ChIP-seq peaks in skin, thymocytes, and brain intersecting with the BUR reference map, indicating that the majority of the peaks are BURs (*Supplementary file 1*). (**D**) The percentage of BURs among all BURs identified in the BUR reference map that intersected with urea SATB1 ChIP-seq peaks identified in skin, thymocytes, and brain. DNase1 HS data are from ENCODE (UW), the CTCF-binding profile and H3K27ac data from standard ChIP-seq are from ENCODE (LICR).

35.0% (based on average of two replicas/cell type), respectively (*Figure 6D* and *Supplementary file 2*). These results indicate that frequency of BURs targeted by SATB1 in vivo can vary more than 10-fold depending on cell type. Interestingly, while the number of BURs bound by SATB1 greatly differs between cell types, the majority (83.9%) of BURs targeted by SATB1 in skin were bound to SATB1 in thymocytes, and nearly all (91.6%) SATB1-bound BURs in thymocytes were bound to SATB1 in brain. Therefore, each cell type has an overlapping set of SATB1-bound BURs in common, with increasing/decreasing numbers of sites depending on cell type, rather than distinct subsets. Despite the highest BUR occupancy by SATB1 being observed in the brain, this is not due to the brain expressing the highest levels of SATB1 protein. In fact, the thymus expresses by far the highest levels of SATB1, which comprises ~0.3% of the total protein (*Dickinson and Kohwi-Shigematsu, 1995*).

## BURs are enriched in LADs

LADs are large mostly heterochromatic domains in close contact with the nuclear lamina and are known to be enriched in AT-rich sequences (*Meuleman et al., 2013*). Although AT-rich sequences are not necessarily BURs, BURs have a minimum of 65% AT content (many of them have a higher AT content) with the ATC sequence context with an exceptionally strong unwinding propensity (*Kohwi-Shigematsu et al., 2013*). We therefore asked whether and to what extent any of the ~240,000 BURs identified in the genome could be found within LADs. We first visually compared the distribution of all potential BURs with LADs mapped by DamID (*Chen et al., 2018*; *Vogel et al., 2007*; *Reddy and Wong, 2025*) and noted some correlation of BURs and LADs. As an example, the 9 Mb region in chromosome 5 containing the neuronal gene cluster locus that encodes subunits of $GABA_A$ receptor complex (*Gabrg1*, and *Gabra1*, *Gaba6*, *Gabrb2*) has a BUR-enriched region overlapping with LADs obtained from thymocytes (*Figure 7A*). We checked if this LAD-BUR overlap is not restricted to this specific region by examining a larger view of a randomly chosen region covering 47.4 Mb in chromosome 3 (*Figure 7—figure supplement 1*). In this region, CTCF-binding sites, DNase1 HS, and standard SATB1 ChIP-seq peaks all converge in the gene-rich inter-LAD regions, whereas the distribution of LADs and BURs largely overlaps, and these regions are enriched in SATB1 urea ChIP-seq peaks.

We then examined the relationship between BUR and LAD distribution in mouse thymocytes at the genome-wide scale. We compared the genomic proximity to lamin B1 (as measured by Dam), SATB1-bound genomic sites (from standard ChIP-seq), SATB1-bound BURs (from urea ChIP-seq) over LADs and flanking regions (±10% outside of the LADs). We found that the genome-wide distribution of LADs coincides with the majority of SATB1-bound BURs, but it is mutually exclusive with SATB1-bound genomic sites that are mapped by standard ChIP-seq, which avoid LADs (*Figure 7B*). In agreement with the result shown in *Figure 2A*, CTCF-bound sites mapped by urea-ChIP-seq coincide with those mapped by standard ChIP-seq and are found at LAD borders and inter-LAD regions (*Figure 7C*), as previously described (*Guelen et al., 2008*). Collectively, these analyses confirmed that the majority of potential BURs from the BUR reference map as well as directly SATB1-bound BURs overlap with LADs (*Figure 7B and C*), whereas CTCF-binding sites avoid LAD-BUR regions and are confined to inter-LADs and LAD boundaries (*Figure 7C*). We conclude that, at the genome-wide scale, the majority of BURs overlap with LADs, suggesting that BURs represent an important sequence component of LADs, some of which are specifically targeted by SATB1. This agrees with earlier findings that LADs are enriched in AT sequences as many BURs are AT-rich (>65% AT) and further identifies LADs as specifically enriched in BURs.

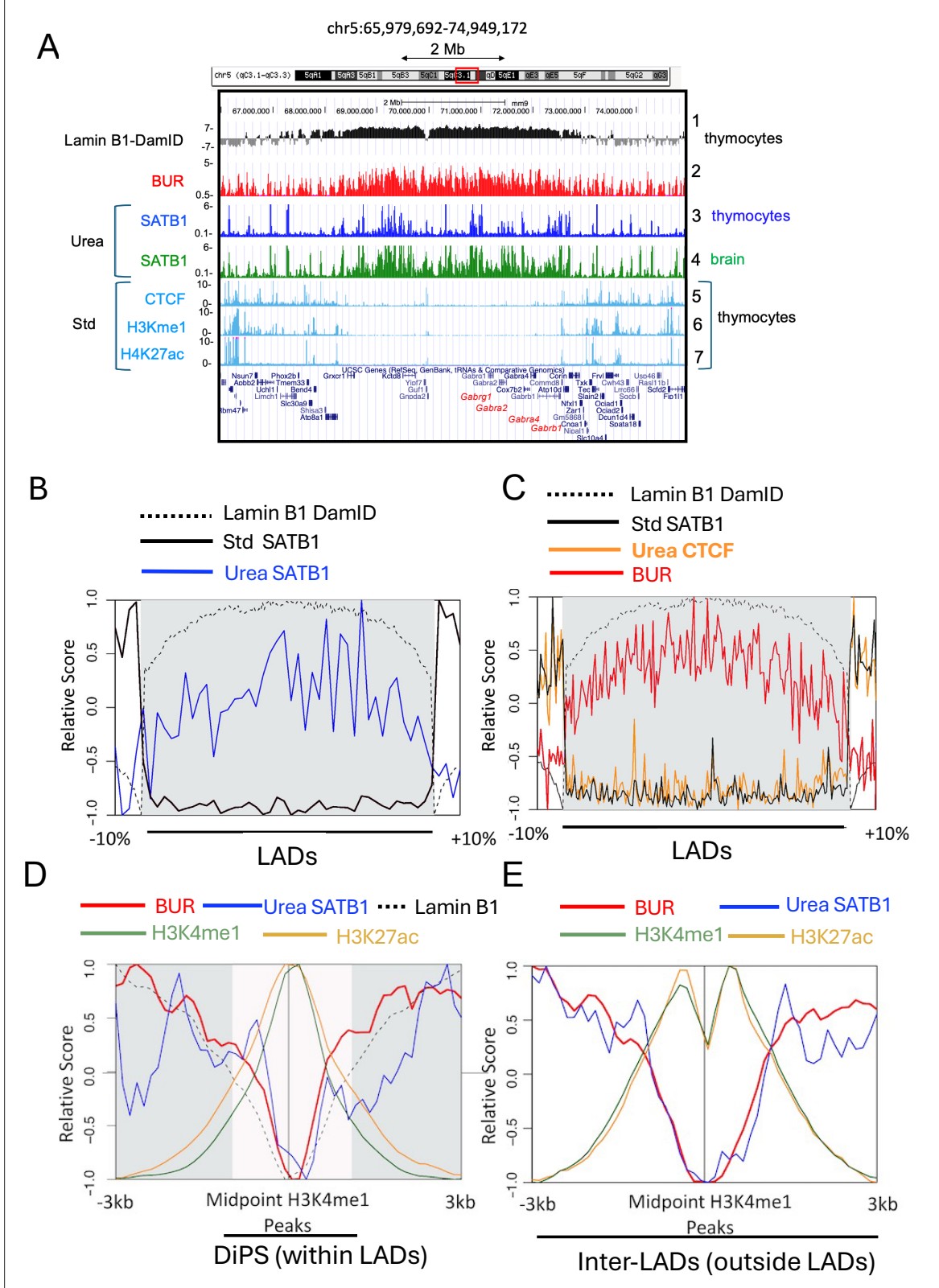

**Figure 7.** BURs largely co-map with LADs. (**A**) The BUR distribution (BUR reference map, track 2) was compared against Lamin B1-DAM-ID sites (*Chen et al., 2018*) (track 1) and SATB1-binding sites derived from urea-ChIP-seq (Urea) (tracks 3 and 4) over an 8.97 Mb region in chromosome 5 containing neuronal genes, *Gabrg1*, *Gabra2*, *Gabra4*, and *Gabrb1*. In this region, co-mapping of BURs and Lamin B1-DAM-ID regions is confirmed. Among BURs, SATB1-bound BURs derived from urea ChIP-seq (Urea) show differences in the number of peaks depending on cell type (see *Supplementary files 1*

*Figure 7 continued on next page*

*Figure 7 continued*

*and 2*). SATB1-binding sites derived from urea ChIP-seq avoid CTCF-binding sites, H3Kme1, H3K27ac sites (from Encode LICR). (**B**) Average Lamin B1 DamID signal (black dotted), standard ChIP signal for SATB1 (black) and urea ChIP for SATB1 (blue) signal over LADs (gray shaded) and flanking regions (inter-LAD sized at 10% of adjacent LAD). (**C**) Average Lamin B1 DamID signal (dotted black), BUR signal (red), and urea ChIP for CTCF (orange) signal over LADs (gray shaded) and flanking regions (inter-LAD sized at 10% of adjacent LAD). (**D**) Average BUR signal (red), urea ChIP for SATB1 (blue) signal, Lamin B1 DamID (black, dotted), H3K4me1 (light green), and H3K27ac (light orange) signals over a ±3 kb window centered at H3K4me1 peaks within LADs (LADs are gray shaded and DiPS that reside inside large LADs are unshaded, see text). (**E**) Average BUR signal (red), urea ChIP for SATB1 (blue) signal, H3K4me1 (light green) and H3K27ac (light orange) signals over a ±3 kb window centered at H3K4me1 peaks outside LADs. Relative distance, projection, and Jaccard tests all indicate strong correlation (p<0.05) between urea ChIP-seq peaks for SATB1 and LADs, between BURs and LADs, but no overlap (p<0.05) between urea ChIP-seq SATB1 and H3K27ac and H3K4me3 peaks. The results in B to E represent normalized average intensities of signals.

The online version of this article includes the following figure supplement(s) for figure 7:

**Figure supplement 1.** BURs and BUR-bound SATB1 largely co-map with LaminB1-DamID.

**Figure supplement 2.** Some SATB1-bound BURs located in SATB1-dependent gene loci are outside of LADs.

**Figure supplement 3.** SATB1 occupancy relative to BURs, and BUR signal relative to SATB1-bound loci.

**Figure supplement 4.** Interaction sites from BUR-1 or BUR-2 are mostly confined to the inter-LAD region.

---

Within larger LAD domains identified, there are smaller regions (<10 kb) that display localized looping away from the lamina (*Wen et al., 2012*; *Leemans et al., 2019*; *Luperchio et al., 2017*). These DiPS (Depleted in Peripheral Signal) display all the hallmarks of poised or active promoters or enhancers and are flanked by CTCF and cohesin. Thus, these small regions embedded in larger LADs are akin to small inter-LADs that are constrained at the lamina due to their short lengths. We next asked if SATB1 binds BURs in these active regions within LAD (DiPS). We examined the distribution of average signals of SATB1-bound BURs relative to enhancers in DiPS marked by H3K4me1 and H3K27ac modifications over a ±3 kb window centered at H3K4me1 peaks. The peaks of enhancer histone marks (H3K27ac and H3K4me1) corresponded to minimum occupancies of SATB1 in DiPS and also excluded BURs, indicating that enhancers are largely excluded from BURs and SATB1-directly bound BURs in DiPS (*Figure 7D*).

We also examined SATB1-bound BURs outside LADs (inter-LADs) in thymocytes. As a specific example, we showed SATB1-bound BUR distribution in close vicinity to genes in inter-LADs containing *Rag1/Rag2* (this region was used for urea 4C-seq analyses above) as well as two other SATB1-regulated loci: *Foxp3* and *Runx3* relative to nearby LADs (*Figure 7—figure supplement 2*). We found that 18% of SATB1-bound BURs reside in inter-LADs, with 82% found within LADs. Among the SATB1-bound BURs in inter-LADs, 39% or 9.5% of them are located close (within ± 5 kb) to the peaks of H3K4me1 or H3K27ac, respectively. Importantly, however, genome-wide analyses revealed that in inter-LAD regions, as well as in DiPS (locally euchromatic regions within LADs), SATB1-bound BURs generally exclude the peaks of enhancers marked by H3K4me1 and H3K27ac (*Figure 7D and E*). This observation is consistent with the fact that these BURs are located in regions of inaccessible chromatin. We viewed relative occupancies of BURs and SATB1-bound sites in LADs and inter-LADs by heatmap (*Figure 7—figure supplement 3*). Consistent with the results described above, SATB1 occupancy is restricted to BUR sites, a fraction of BURs is targeted by SATB1 in both inter-LADs and LADs (the former being a much fewer number of BURs). We also examined the distribution of urea 4C-detected BUR-1 or BUR-2 interactions relative to LADs. These SATB1-mediated interactions were found to be well constrained within inter-LADs enriched in histone marks for enhancers (*Figure 7—figure supplement 4*). This remains the case even for sites with much longer interactions, as these sites are located in accessible regions between LADs, confirmed by viewing individual sites at high resolution. These results demonstrate that the inter-LAD BURs (BUR-1 and BUR-2), bound to SATB1 nuclear substructure, are contacting accessible chromatin at numerous locations over long distances. Taken together, these data demonstrate that SATB1-bound BURs reside in locally inaccessible chromatin whether they are found inside repressive LADs or in DiPS and in regions within inter-LADs that are otherwise marked as active chromatin.

## Discussion

Here, we used a systematic approach to compare and interrogate the chromatin binding profiles of a genome organizing protein SATB1 detected by two alternative ChIP-seq approaches. Critically, by establishing a urea purification-based ChIP-seq method that enriches for direct DNA-binding events, we found that SATB1 directly binds BURs genome-wide. This is in stark contrast to previously identified SATB1 binding to enhancers and promoters in open chromatin by standard ChIP-seq. Nevertheless, the two contrasting DNA-binding profiles for SATB1 are both valid, requiring SATB1 for the ChIP-seq peaks. These differing SATB1-binding profiles suggest that SATB1 is engaged in chromatin organization through both direct binding to BURs, anchoring them to the subnuclear SATB1 protein meshwork, as well as by indirect binding to the DNase 1-accessible (active) chromatin region, enriched in genes and regulatory regions. We show that SATB1 is required for megabase-level interactions, connecting select BURs to 'open' chromatin, and that such interactions are linked with cell-type-specific gene expression regulated by SATB1 in thymocytes.

Using urea ChIP-seq, which stringently purifies chromatin, we identified a new type of chromatin organization formed by direct binding of SATB1 with BURs, genomic elements distributed across the mouse genome. While this SATB1-mediated chromatin organization contrasts with previous data from standard ChIP-seq or CUT&Tag assays, which indicate SATB1 mainly interacts with enhancers and promoters located within the 'open chromatin' regions, our data agree with even earlier in vitro findings that SATB1 binds to AT-rich BURs. Thus, standard ChIP-seq (or Cut&Tag) and urea ChIP-seq enrich different and almost mutually exclusive SATB1 profiles, suggesting that these approaches are interrogating the genome in different ways. It is important to note that while urea ChIP-seq and standard ChIP-seq for SATB1 showed very different profiles, the binding profiles of CTCF and other proteins did not differ between the two methods. This is consistent with SATB1 interacting with chromatin in a unique way by forming a subnuclear proteinaceous network anchoring a subset of BURs (*Cai et al., 2003*).

## Differences between standard and urea ChIP-seq

In order to understand why chromatin binding proteins might show different binding modalities in urea ChIP-seq versus more standard approaches, it is important to understand the differences in the protocols. Standard ChIP-seq protocols use crosslinked whole cells or nuclei without purification of crosslinked chromatin, and immunoprecipitation is typically performed with the 'cleared' suspension after extracts of sonicated crosslinked chromatin are centrifuged, thus removing most of the insoluble material. Even if this fraction is not cleared in this way, the insoluble material does not perform well in immunoprecipitation assays (*Gesson et al., 2016*). Previous studies have identified SATB1 as a salt-extraction resistant (nuclear matrix) protein (*Cai et al., 2003*; *de Belle et al., 1998*). Thus, using canonical ChIP-seq approaches, SATB1-directly bound BURs in the DNase 1-inaccessible nuclear substructure would necessarily be lost in the insoluble fraction. Conversely, any SATB1-associated genomic regions in open chromatin regions (direct or indirect) would become the predominantly captured fraction that is amplified during PCR-based library preparations. Thus, SATB1 directly bound to BURs in the high-salt extraction resistant nuclear substructure is likely missed by standard ChIP-seq (*Feng et al., 2022*; *Zelenka et al., 2022*) and by CUT&Tag (*Wang et al., 2023*), which is also known to preferentially detect protein interactions in open chromatin regions. In contrast, in urea ChIP-seq, whole, intact crosslinked chromatin is purified through urea ultracentrifugation, first removing all indirectly bound proteins from chromatin. Subsequently, the entire urea-purified crosslinked chromatin is quantitatively solubilized. The two main advantages of urea ChIP-seq are that (1) only direct-binding profiles will be obtained for a protein of interest, and (2) chromatin tightly associated with nuclear substructures, which are typically discarded in the insoluble fractions in standard ChIP-seq, are retained as whole intact crosslinked genomic DNA in urea ChIP-seq. Therefore, all direct DNA binding profiles can be obtained, regardless of the original chromatin state (i.e. no bias toward open chromatin). Thus, urea ChIP-seq is uniquely suited to detect direct binding sites of SATB1 as well as other BUR-binding proteins genome-wide. For non-BUR-binding proteins (e.g. CTCF and PRC2 core subunits), urea ChIP-seq generates identical DNA binding profiles as standard ChIP-seq. Although we detected some 'phantom' or false-positive peaks using SATB1-KO cells by standard ChIP-seq (*Supplementary files 3 and 4*), under our experimental conditions, we confirmed that the majority of SATB1-binding peaks are SATB1-dependent. Importantly, these SATB1 peaks from standard ChIP-seq were essentially

undetected by urea ChIP-seq, suggesting that SATB1 indirectly binds to these sites presumably by associating with nuclear proteins bound to their target sites in 'open' chromatin. SATB1 is equipped with domains responsible for specific binding to BURs (*Galande et al., 2001*; *Dickinson et al., 1997*; *Nakagomi et al., 1994*; *Wang et al., 2014*) as well as for associating with or recruiting chromatin binding proteins to specific genomic sites (e.g. CHRAC and ACF1 nucleosome mobilizing complex subunits, Brg1, p300, Hdac1, Gata3, Stat6, c-Maf, Pit1, and β-catenin) (*Cai et al., 2003*; *Cai et al., 2006*; *Skowronska-Krawczyk et al., 2014*; *Han et al., 2008*; *Pavan Kumar et al., 2006*; *Notani et al., 2011*; *Stephen et al., 2017*; *Yasui et al., 2002*; *Notani et al., 2010*). Therefore, we anticipate that SATB1 binds indirectly and dynamically to regulatory regions in open chromatin by associating with many chromatin-modifying and transcription factors to form protein complexes to support tissue-specific gene expression. Despite the contrasting SATB1-binding profiles obtained from standard ChIP-seq and urea ChIP-seq, generation of the peaks from both methods requires SATB1. Therefore, SATB1-mediated chromatin organization appears to have two components: one involves direct binding to BURs, while the other involves indirect binding to other chromatin-interacting proteins at genomic sites in accessible chromatin.

## Two-tiered chromatin organization mediated by SATB1

Considering that the two contrasting SATB1-binding profiles require SATB1, an important question emerges as to whether there is a functional or structural link between the two SATB1-bound chromatin regions. We conceive that SATB1 might have a role in bringing together chromatin structure formed by BURs tightly bound to nuclear substructure and a highly accessible 'open' chromatin enriched in transcription-related proteins. Our urea 4C-seq results suggest that these two regions are connected via SATB1-mediated interactions involving BURs. The ligation step in urea 4C-seq enabled us to capture chromatin sites that were in close spatial proximity to the bait site (e.g. a BUR) at the time of crosslinking. Apparently, 3D structures fixed by formaldehyde retained sufficient chromatin proximity after urea purification to enable capture of BUR-interacting fragments. The urea 4C-seq revealed that a SATB1-bound BUR site (BUR-1 as a bait), located at the border of a gene desert and a gene-rich region containing SATB1-regulated genes (*Rag 1* and *Rag2*), interacts extensively over a 5.8 Mb gene-rich region with many gene loci, such as enhancers, promoters, BURs, and intergenic regions. Such ultra-long-distance interactions from BUR-1 predominantly cover the entire 5.8 Mb gene-rich region, mostly avoiding the gene desert region. Such interactions were confirmed with another bait (BUR-2).

The SATB1-mediated chromatin interactions from BUR-1 and BUR-2 (validation control) as baits, covering the 5.8 Mb *Rag1/Rag2*-containing gene-rich region, are tightly linked to SATB1-dependent expression of three genes, *Rag1, Rag2 and Prrg4* (adjusted p value<0.05) out of 49 genes in this region in thymocytes. BUR-1 further interacts with far distal gene-rich regions in cis in chromosome 2 (>44 Mb), skipping interstitial gene desert regions in a SATB1-dependent manner, albeit through weaker interactions (*Figure 5—figure supplement 3*). Within these distal gene-rich regions, 10 additional SATB1-dependent genes were identified in thymocytes. These long-distance chromatin interactions are at a much greater scale than those previously observed, such as T helper 2 cell activation induced dense chromatin looping in the 200 kb cytokine gene cluster, including several BURs and regulatory regions, tightly linked to activation of the cytokine genes (*Cai et al., 2006*) and interactions at the subTAD levels (or <~500 Kb) between specific enhancers or enhancers to promoters in thymocytes (*Feng et al., 2022*; *Zelenka et al., 2022*; *Wang et al., 2023*). Combining results from SATB1-binding profiles derived from urea ChIP-seq and standard ChIP-seq as well as urea 4C-seq, a model is illustrated how the two chromatin regions might interact depending on SATB1 (*Figure 8*). In this model, the stable SATB1 binding to BURs in the SATB1-rich nuclear substructure forms a foundational chromatin scaffold and SATB1 tethers highly accessible 'open' chromatin regions to this scaffold by binding to regulatory protein complexes—the SATB1 proteinaceous matrix would serve as a scaffold and interface. We anticipate that the latter interactions will be dynamic and form the highly interacting chromatin architecture over gene-rich accessible chromatin regions, bringing regulatory sequences in close vicinities. Both components of SATB1-mediated chromatin organization likely underlie regulation of gene expression. This two-tier model of SATB1 chromatin organization is highly reminiscent of previously published results on SATB1's role in transcription factor Pit1-regulated gene expression in pituitary glands (*Skowronska-Krawczyk et al., 2014*). SATB1 associates with Pit1 and β-catenin, and SATB1 is required for tethering Pit1-bound enhancers to the subnuclear architectural

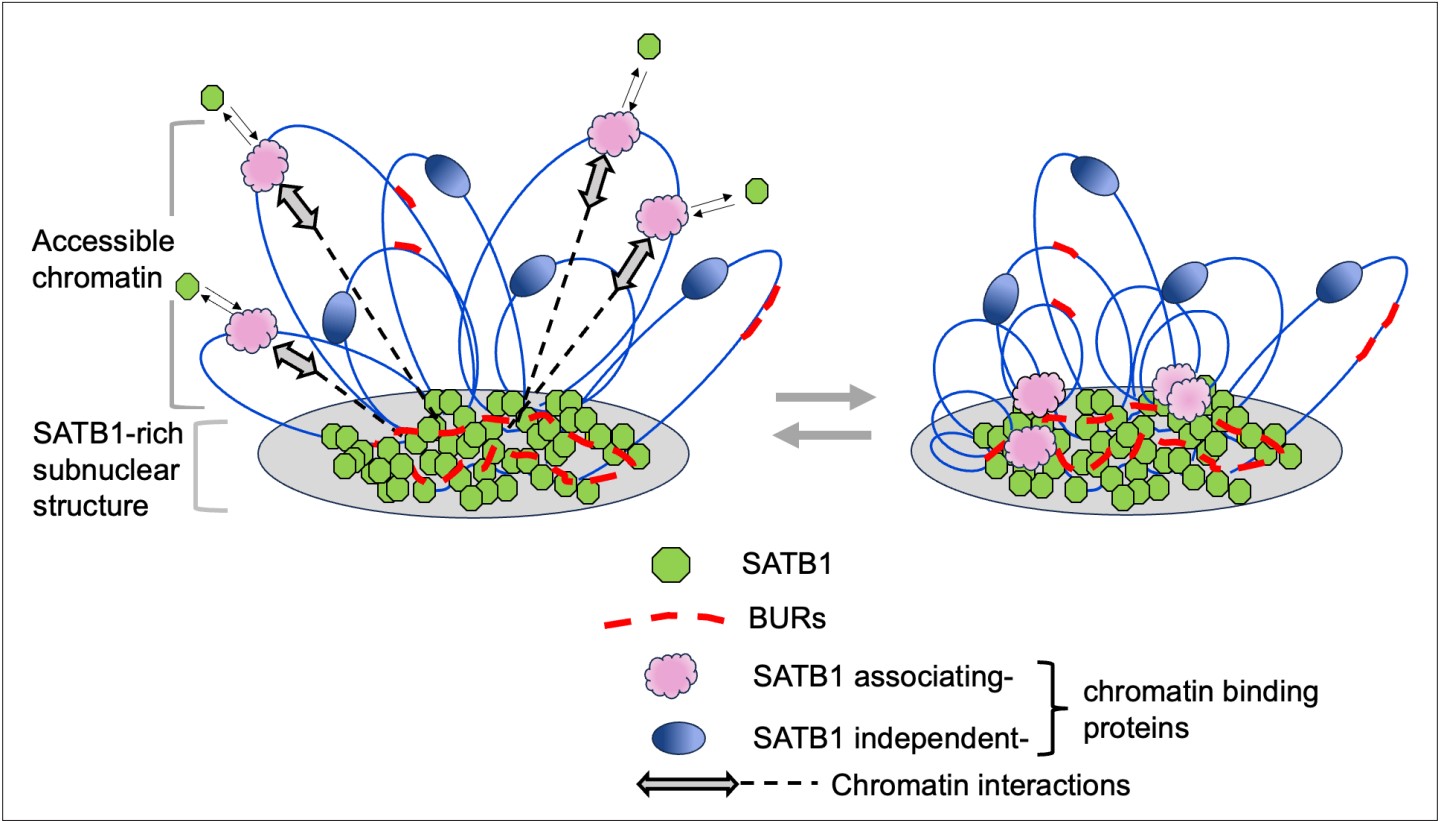

**Figure 8.** A model for SATB1-mediated chromatin organization with direct and indirect binding of SATB1 with chromatin. This model is based on the chromatin-binding profile of SATB1 obtained from urea ChIP-seq, which shows direct binding of SATB1 to BURs, and from standard ChIP-seq, which indicates presumably indirect binding of SATB1 to gene-rich, accessible chromatin regions. Additionally, the model incorporates chromatin interactions of selected SATB1-bound BURs detected by urea 4C-seq in this study. Once BURs are bound by SATB1, they are tightly anchored to the SATB1-rich nuclear substructure, which resists high salt extraction, as demonstrated in previous studies (*Cai et al., 2003*; *de Belle et al., 1998*). We showed that SATB1-bound BURs reside in discrete DNase 1-inaccessible chromatin, excluding enhancers and CTCF-bound sites, even in otherwise gene-rich accessible chromatin neighborhoods outside LADs. We propose that the direct binding of SATB1 to BURs within the SATB1-rich nuclear substructure is stable and forms a chromatin scaffold. This chromatin scaffold interacts dynamically with active gene-rich regions through indirect binding, creating functional chromatin complexes (or hubs) that regulate gene expression depending on the cell type. Future studies are needed to determine the potential role of soluble SATB1 proteins and other regulatory factors in these interaction events.

structure that is resistant to salt extraction (note: it is referred to as matrin-3 rich network, as matrin-3 is enriched in such substructure as well). A naturally occurring mutation of human PIT1 (R271W) that causes combined pituitary hormone deficiency fails to associate with SATB1 and β-catenin. Further studies with mutated R271WPit1 revealed that the tethering of Pit1-bound enhancers to the insoluble nuclear substructure is essential for effective activation of the Pit1-regulated transcriptional program.

Additional research will deepen our understanding of how SATB1 regulates cell-type-specific gene expression. The Wnt signaling pathway plays pivotal roles in carcinogenesis and development (*Song et al., 2024*; *Zhan et al., 2017*), and it will be important to investigate the role and nature of SATB1's involvement in this pathway. Beta-catenin is the key component of the canonical Wnt signaling pathway; upon nuclear translocation, it associates with the TCF/LEF family of transcription factors to activate target genes. Because SATB1 also associates with beta-catenin, it is expected that SATB1 may influence target gene expression, similar to its effects on Pit1's target genes. In addition, non-coding RNAs may have an important role in SATB1 chromatin organization and gene expression. Especially, specific long non-coding RNAs (lncRNAs) that remain associated with crosslinked chromatin after urea purification may deserve attention (unpublished results). Most non-coding RNA is associated with RNA-binding proteins (RBP) to form the ribonucleoprotein network which provides RNA structural scaffolds for the 3D chromatin organization and function (*Nickerson et al., 1989*; *Nickerson, 2022*; *Thakur and Henikoff, 2020*; *Michieletto and Gilbert, 2019*). The SATB1 subnuclear architecture has

a prominent cage-like distribution in thymocytes, while neurons have a spider-web-like distribution. It would be of interest to address whether RNA, including lncRNAs, plays a role in the SATB1-rich subnuclear proteinaceous scaffold formation depending on cell type and directing SATB1 to target specific sets of BURs in a cell function-specific manner.

## Lack of overlapping function for SATB1 and CTCF

Results obtained using urea-purified crosslinked chromatin show no relationship between CTCF and SATB1. Several pieces of evidence support this conclusion. First, genome-wide CTCF-binding profiles and Hi-C profiles remained largely unaltered by SATB1 deletion in thymocyte (*Figure 3*), a finding supported by other studies (*Feng et al., 2022*; *Zelenka et al., 2022*). Second, by urea ChIP-seq, SATB1:BUR binding sites precisely exclude CTCF-binding sites, which contrasts with recent findings that SATB1 and CTCF-binding profiles overlap by standard ChIP-seq (*Zelenka et al., 2022*) or by CUT&Tag (*Wang et al., 2023*). Third, chromatin interactions (>5.7 Mb) from a SATB1-bound BUR are not constrained to individual TADs but span multiple TADs. Fourth, by urea ChIP-Western, SATB1 does not interact with CTCF or cohesin on chromatin. Fifth, CTCF does not bind to BURs. Therefore, chromatin organization mediated by SATB1 through direct binding to BURs is distinct from that mediated by CTCF and cohesion.

## Potential biological significance of BURs mostly enriched in LADs

We identify BURs as genomic elements enriched in LADs which are large chromatin domains located proximal to the nuclear lamina. LADs and LAD borders have critical roles in gene regulation (e.g. the *Tcrb* locus; *van Steensel and Belmont, 2017*; *Chen et al., 2018*). Except for being generally AT-rich, genomic features of LADs have thus far been poorly characterized. BURs are not simply any AT-rich sequence, but rather, are characterized by a specific ATC sequence context. BURs are important genomic elements that are specifically targeted by SATB1, which has a role in cell-type-specific gene expression (*Kohwi-Shigematsu et al., 2013*). Thus, discovering BUR enrichment in LADs uncovers mechanistic insights into the biological function of LADs in gene regulation. Although LADs form heterochromatin proximal to the nuclear envelope, LAD organization is dynamic. While LADs are certainly constrained to the nuclear lamina, regions within LADs extend well into the nuclear interior and display dynamic interactions with the lamin proteins. In addition, LADs are dynamic during differentiation or other cell-state changes. For instance, during differentiation of mouse cells, hundreds of genes display dynamic intranuclear spatial repositioning relative to the nuclear lamina (*Peric-Hupkes et al., 2010*). Some LAD regions dynamically contact the nuclear lamina, and gene movement in and out of the nuclear periphery in a circadian rhythm has been reported (*Zhao et al., 2015*). Recently, relocation of the *hunchback* gene in *Drosophila* neuroblasts to nuclear lamina has been shown to require polycomb proteins binding to a specific intronic element of the gene locus. This process is important for heritable gene silencing to regulate neuroblast competence (*Lucas et al., 2021*). The organization of genomic DNA at the lamina has been linked to both inactive chromatin state and the nuclear lamins, along with other nuclear membrane proteins (*van Steensel and Belmont, 2017*; *Solovei et al., 2009*; *Kind et al., 2013*; *Bian et al., 2013*; *Wong et al., 2014*; *Harr et al., 2015*; *Wong et al., 2021*). While chromatin state is likely important for both positioning at and de-localization from the lamina, little information is available concerning in vivo mechanisms specifying which regions within LADs could be targeted for de-localization and subsequent cell-type-specific gene activation. The SATB subnuclear architecture is found mostly in the nuclear interior of thymocytes where many individually cloned BURs have been detected by DNA-FISH. Among such interiorly localized BURs, a BUR 5′ of *Myc* was confirmed to co-localize with SATB1 by immunoDNA-FISH in thymocytes (*Cai et al., 2003*). A closer look at immunostaining profiles of SATB1 distribution in thymocyte nuclei, however, shows that a small reproducible fraction of SATB1 resides in the peripheral zone of nuclei and in some cells the tip of the SATB1 network extends to and touches this region of the nucleus (*Cai et al., 2003*).

Future studies are needed to investigate the spatial context of a subset of BURs and LADs in relation to SATB1 subnuclear architecture, depending on the cell type. For instance, many SATB1-bound BURs are identified in the *Gabrg1* and *Gabra2* loci in neurons, but not in thymocytes (*Figure 6B*), even though thymocytes express SATB1 at high levels. Therefore, studies on nuclear positioning of select BURs in neurons, thymocytes, and in non-SATB1-expressing cells would provide valuable insights. In

addition to LADs, transcriptionally active gene-rich regions outside LADs also contain BURs, albeit at a considerably lower density compared to LADs (e.g. *Rag1/Rag2, Foxp3,* and *Runx3* loci). Some of these BURs located near enhancers are bound by SATB1. It is of interest that the number of BURs targeted by SATB1 greatly varies depending on cell type, where neurons have by far the greatest number of BURs captured by SATB1. This may reflect the level of plasticity for gene transcription or number of genes that must be dynamically regulated in each cell type in which SATB1 is expressed. There is growing evidence supporting the critical roles of chromatin topology and nuclear architecture in neuronal response to external stimuli to establish precise expression programs associated with cognition (*Watson and Tsai, 2017*). In fact, SATB1 has roles in early postnatal cortical neuron development and neuronal connectivity (*Balamotis et al., 2012*; *Close et al., 2012*). The exceptionally large number of BURs directly interacting with SATB1 in cortical neurons may reflect specialized function of neurons that might require highly dynamic transcriptional regulation to enable rapid response to a wide variety of environmental stimuli. How SATB1 selects specific subsets of BURs and if and how SATB1's selection of BURs contributes to cell-type-specific gene expression are important topics for future research.

## A broader significance of BUR-binding proteins beyond SATB1

Importantly, SATB1 is not the only BUR-binding protein, indicating the broader biological significance of BURs as genomic elements. Although SATB1 confers by far the highest affinity to BURs, several other nuclear factors (e.g. nucleolin, HMGI/Y, PARP1, Ku70/86 heterodimer) have also been verified to directly and specifically bind BURs in vitro (*Dickinson and Kohwi-Shigematsu, 1995*; *Galande and Kohwi-Shigematsu, 1999*; *Liu et al., 1999*). Recently, we identified Tip5/Baz2a as an additional strong BUR-direct binding protein from mouse embryonic stem cells (mESCs) that is essential for pluripotency (submitted for publication). Similar to SATB1, this nuclear protein also binds specifically to BURs in mESCs, and identification of BURs as its direct targets in mESCs requires urea ChIP-seq as well. Despite the identical BUR binding specificity of SATB1 and this stem cell factor, these two proteins confer entirely different functions in distinct cell types. Therefore, BURs may serve as core genomic marks that can be selectively targeted by distinct BUR-binding proteins to form unique 3D chromatin regulatory network depending on the BUR-binding protein for specific cell functions. BURs in the mammalian genome may have similarities in function with the 'tethering elements' in *Drosophila* (*Batut et al., 2022*; *Levo et al., 2022*). These elements can be bound by regulators such as pioneering factors and polycomb proteins and were shown to promote interactions of regulatory regions up to ~250 kb within TADs to regulate transcription of genes. Most recently, meta-loops constituting a meta-domain spanning numerous TADs in the CNS of *Drosophila* were reported (*Mohana et al., 2023*; *Lizana and Schwartz, 2024*). Although meta-loops are similar in size with SATB1-mediated chromatin interactions, the CNS meta-loops have different features. They are formed by transcription factors (e.g. CTCF and GAF), half of them is anchored at the TAD borders, and the metadomain boundaries are CTCF-bound. It appears that clusters of very long chromatin looping events linked to gene expression may form by multiple mechanisms across species. Future research on BUR-mediated genome organization regulated by distinct BUR-binding proteins will likely provide more in-depth understanding about cell-type-specific mechanisms driving 3D chromatin architecture that contributes to major changes in gene expression and cell phenotypes.

# Materials and methods

## Key resources table

| Reagent type (species) or resource | Designation | Source or reference | Identifiers | Additional information |
|---|---|---|---|---|
| Genetic reagent (Mouse) | *Satb1*<sup>fl/fl</sup> | This paper Generated by Kohwi's lab; details are in Balamotis et al, manuscript in preparation | *Satb1cKO* | Maintained on a C57BL/6J background |

*Continued on next page*

*Continued*

| Reagent type (species) or resource | Designation | Source or reference | Identifiers | Additional information |
|---|---|---|---|---|
| Genetic reagent (Mouse) | *Satb1*$^{fl/fl}$: *Cd4cre* | ***Kakugawa et al., 2017***; ***Kitagawa et al., 2017*** | *Satb1*$^{fl/fl}$: *Cd4cre* | Maintained on a C57BL/6J background |
| Genetic reagent (Mouse) | *Satb1*$^{-/-}$ | ***Alvarez et al., 2000***; Generated by Kohwi's lab | *Satb1*KO | Maintained on a C57BL/6J background |
| Genetic reagent (Mouse) | *Satb1*$^{+/+}$ | The Jackson Laboratory | WT | C57BL/6J Strain #:000664 RRID:IMSR_JAX:000664 |
| Antibody | Anti-CTCF (Rabbit polyclonal) | Active MOTIF | Catalog # 61311 | ChIP |
| Antibody | Anti-CTCF (Rabbit monoclonal) | Abcam | Catalog # Ab70303 | Western (1:1000) |
| Antibody | Anti-CTCF (Rabbit polyclonal) | Millipore | Catalog # 7–729 | ChIP Western (1:1000) |
| Antibody | Anti-RAD21 (Rabbit polyclonal) | ABclonal Science | Catalog # A18850 | Western (1:1000) |
| Antibody | Anti-SATB1 (Rabbit polyclonal) | Generated by Kohwi's Lab | 1583D | ChIP (after SATB2 absorption) |
| Antibody | Anti-SATB1 (Rabbit monoclonal) | Abcam | Catalog # Ab109122 | ChIP Western (1:1000) |
| Antibody | Anti-Histone H3 (Rabbit polyclonal) | Abcam | Catalog # Ab1791 | Western (1:4000) |
| Chemical compound, drug; | Proteinase inhibitor; cOmplete Tablets | Roche | Catalog # 4693132001 | |
| Chemical compound, drug; | Urea | Sigma | Catalog # U5378-500G | |
| Other | RNase inhibitor | New England Biolabs | Catalog # M030L | |
| Other | RNaseA | ThermoFisher | Catalog # EN0531 | |
| Other | AmpliTaq Gold DNA polymerase | AppliedBiosystems | Catalog # 4311806 | |
| Other | Dynabeads Protein A (protein purification beads) | Invitrogen | Catalog # 10001D | |
| Other | Dynabeads Protein A/G (protein purification beads) | ThermoFisher | Catalog # 26,157 X | |
| Commercial assay or kit | Quick PCR purification kit | Invitrogen (Qiagen) | Catalog # K310001 | |
| Commercial assay or kit | Quick PCR purification kit | Macherey-Nagel (MN) | Catalog # 740609.25 | |
| Commercial assay or kit | TruPLEX DNA-seq kit | Rubicon Genomics | Catalog # QAM-150–001 | |
| Commercial assay or kit | NEBNext Ultra II DNA Library Prep Kit for Illumina | New England Biolab | Catalog # NEB #71035 L | |
| Commercial assay or kit | Arima HiC +kit | Arima Genomics | Catalog # A510008 | |

*Continued on next page*

*Continued*

| Reagent type (species) or resource | Designation | Source or reference | Identifiers | Additional information |
|---|---|---|---|---|
| Other | SPRIselect (DNA purification beads) | Beckman Coulter | Catalog # B23317 | |
| Sequence-based reagent | Linker top strand (Biotin modified) | This paper | Linker oligomer | aattcggtacctctagagat*at* ccgatcgctcgagaagctt<br>*: Biotin modified<br>5′→3′ |
| Sequence-based reagent | Linker bottom strand | This paper | Linker oligomer | aagcttctcgagcgatcggatatc tctagaggtaccgaatt<br>5′→3′ |
| Sequence-based reagent | Illumina sequence-BUR-1-bait oligomer | This paper | Sequence primer | AATGATACGGCGACCAC CGAGATCTAC ACTCTTTCCCTACACGAC-gaagggggtaggaaagaaagctt<br>Capital letter: Illumina sequence<br>5′→3′ |
| Sequence-based reagent | Illumina sequence-BUR-2-bait oligomer | This paper | Sequence primer | AATGATACGGCGACCAC CGAGATCTAC ACTCTTTCCCTACACGAC-cgactcattcctactgaagctt<br>Capital letter: Illumina sequence<br>5′→3′ |
| Sequence-based reagent | Illumina sequence-linker oligomer | This paper | Sequence primer | CAAGCAGAAGACGGCATACGAGAT- (barcode, e.g. CGTGAT)-GTGACTGGAGTTC-gcgatcggatatctctagagg taccgaattc<br>Capital letter: Illumina sequence<br>5′→3′ |
| Sequence-based reagent | BUR-1-bait sequence oligomer: | This paper | Sequence primer | CTTTCCCTACACGAC-gaagggggtaggaaagaaagctt<br>Capital letter: Illumina sequence<br>5′→3′ |
| Sequence-based reagent | BUR-2-bait sequence oligomer | This paper | Sequence primer | ACTCTTTCCCTACACGAC-cgactcattcctactgaagctt<br>Capital letter: Illumina sequence<br>5′→3′ |
| Sequence-based reagent | Linker specific barcode oligomer | This paper | Sequence primer | gaattcggtacctctagagatatccgatcgc-GAACTCCAGTCAC<br>Capital letter: Illumina sequence<br>5′→3′ |
| Sequence-based reagent | *Satb1fl/fl* mice genotyping validation CKO2SETF (forward) | This paper | PCR primer CKO2SETF | ACGCAAACAGAACCCACTG<br>5′→3′ |
| Sequence-based reagent | *Satb1fl/fl* mice genotyping validation CKO2SETR (reverse) | This paper | PCR primer CKO2SETR | ACCAGGCAGAAAAATCATTG<br>5′→3′ |

## Mice

*Satb1fl/fl* mice were generated in Kohwi-Shigematsu's lab, and the detailed methods for their production are described (Balamotis et al., manuscript in preparation). Briefly, bacterial artificial chromosome BAC RP21-130D1, which contains the *Satb1* locus, was used as the starting material for cloning and recombination. The final product, *Satb1fl/fl* mice, has LoxP recombination sites flanking *Satb1* exons 3, 4, and 5 on both alleles. In the presence of Cre recombinase, exons 3, 4, and 5 will be removed. It is important to note that the first 70 amino acids of the SATB1 protein are correctly translated: the last four amino acids are NLRK. A frameshift occurs after these 70 amino acids, resulting in a different sequence of 38 amino acids that ends with YDG before reaching a stop codon. The oligonucleotide primers for validating the recombination events are [CKO2SETF (forward), 5′-ACGCAAAC AGAACCCACTG-3′; CKO2SETR (reverse), 5′-ACCAGGCAGAAAAATCATTG-3′], which are designed to flank a single LoxP site located between exon 2 and exon 3. The PCR product containing the LoxP site is 200 bp in size, and after the LoxP deletion by the Cre recombinase, it will be reduced

to 124 bp. *Satb1^fl/fl* mice are maintained on the C57BL/6 background. *Satb1^fl/fl:Cd4-Cre* mice were generated in I. Taniuchi's lab (Riken, Japan) and S. Sakaguchi's lab (The University of Osaka, Japan) by crossing *Satb1^fl/fl* mice with *Cd4-Cre* mice as described **Kakugawa et al., 2017**; **Kitagawa et al., 2017**. The generation and phenotypes of *Satb1^-/-* mice are published (**Alvarez et al., 2000**). All the work involving animals was approved by the University of California San Francisco Institutional Animal Care and Use Committee, with the IACUC approval number AN206117.

## Primary cells

Thymi were isolated from 2-week-old *Satb1^+/+* and *Satb1^-/-* mice, as well as from *Satb1^+/+:Cd4-Cre* and *Satb1^fl/fl:Cd4-Cre* mice. The brain (frontal cortex) was isolated from 6- to 10-week-old *Satb1^+/+* mice, and skin was isolated from postnatal day 0.5 of C57BL/6J mice. Primary epidermal keratinocytes were isolated from the skin as described (**Poterlowicz et al., 2017**). Both thymi and frontal cortexes were crushed, and cells were filtered through a 70 μm cell strainer, pelleted by centrifugation, and re-suspended in pre-chilled 1xPBS buffer containing 2% fetal calf serum. After the cells were washed twice with 1xPBS (with 2% fetal calf serum) and resuspended in the same buffer, these cells were filtered once again through a 70 μm cell strainer and washed twice.

## Antibodies

Antibodies used and for their purposes are the following. CTCF for ChIP-seq: Active MOTIF (61311, lot# 34614003), CTCF for Western: Abcam (ab70303) and Millipore (07–729; lot#–3429114), RAD21 for ChIP-seq and Western: ABclonal (A18850, lot# 3521809001), SATB1 for ChIP-seq and Western: rabbit polyclonal 1583D prepared by Kohwi-Shigematsu's laboratory and Abcam (ab109122, lot#GR137720-13) and Histone H3 for Western: Abcam (ab1791, lotGR3236305-2). Anti-SATB1 antibody (1583D) was used after SATB2 absorption.

## Urea-purified chromatin immunoprecipitation and urea ChIP-seq

Thymocytes and single-cell suspension prepared for cortex and skin were cross-linked with 1% formaldehyde for 10 min at room temperature, followed by addition of glycine at 0.125 M for 5 min to quench cross-linking reaction. These cells were washed with 1 X PBS, suspended in 1% BSA-PBS, and cells were precipitated in a tube. These cells were re-suspended in lysis buffer (4% SDS, 50 mM Tris-HCl [pH 8.0], 10 mM EDTA, 100 mM NaCl) with Protease inhibitor (Roche) and RNase inhibitor. The resulting cell lysate was incubated for 30 min at room temperature and loaded on top of 8 M urea followed by ultracentrifugation in a Beckman TL-100 Ultracentrifuge for 4–5 hrs at 45 K rpm (181,000 g, g=relative centrifugation force or RCF; TLS55 rotor). Recovered chromatin pellet was sonicated to generate fragment size ranging 300–1000 bp using Branson sonifier (20% power, 10 s on and 30 s off for 4–6 cycles) in 300 μl of sonication buffer [0.5% Na-laurylsarcosine, 0.1% deoxycholate (DOC), 1 mM EDTA, 0.5 mM EGTA, 10 mM TrisHCl, 100 mM NaCl] with Protease inhibitor and RNase inhibitor. Note: urea-purified crosslinked chromatin is highly vulnerable to sonication; therefore, it needs to adjust the sonication condition depending on cell type and/or the size of the precipitate DNA. The sonicated chromatin with 1% Triton X-100 was dialyzed against dialysis buffer (10 mM Tris-HCl [pH 8.0], 100 mM NaCl, 1 mM EDTA, 5% glycerol) and dialyzed for 16 hr at 4 °C (this dialysis is optional). The chromatin sample was centrifuged at 15,000 × g for 5 min at 4 °C to remove a trace amount of residual insoluble components. The test-reverse cross-linking of chromatin aliquot (5–10 μl of the final volume) was done at 65 °C for 2–3 hr in buffer containing 1% SDS, 100 mM NaCl, 1 mM EDTA, 10 mM Tris. Samples were further treated by 100 μg/ml RNase A at 37 °C for 1 hr, 200 μg/ml proteinase K at 60 °C for more than 3 hrs, followed by DNA purification using QIAquick PCR purification kit (Qiagen), and shearing efficiency was examined by Bioanalyzer 2100 (Agilent Technologies Inc). For chromatin-immunoprecipitation, 1 μg of an antibody of choice was added to 10~20 μg of sonicated solubilized chromatin in 350 μl in dilution buffer (1% Triton X-100, 0.01% SDS, 1 mM EDTA, 15 mM Tris-HCl, 100 mM NaCl) and incubated overnight with rotation at 4 °C. At least three chromatin samples were prepared for ChIP-seq from independent experiments to confirm the results. The chromatin fragments bound to antibodies (after incubation overnight) were captured using 20 μl of protein-A, protein-G, or protein AG Dynabeads Invitrogen; pre-washed with 0.1% BSA (Fraction V, Sigma), by 4 hrs incubation time at 4 °C. Dynabeads bound to chromatin fragments were washed twice with Binding buffer (10 mM Tris, pH8.0, 1 mM EDTA, 1 mM EGTA, 0.5% Triton X-100,

0.1% DOC, 100 mM NaCl), twice with LiCl buffer (100 mM Tris [pH 8.0], 1 mM EDTA, 250 mM LiCl, 0.5% NP-40, and 0.5% DOC). The tubes were changed and washed twice with 0.1 M NaCl-TE buffer. The immunoprecipitated samples were eluted from Dynabeads by elution buffer (10 mM Tris-HCl [pH 8.0], 1 mM EDTA, 100 mM NaCl, and 1% SDS) at 65 °C for 20 min. After elution, ChIP DNAs were reverse-crosslinked at 65 °C for 6 hr, followed by RNase-A treatment at 37 °C for 1 hr and proteinase K treatments for 3 hr at 60 °C. The recovered DNA was further purified with a QIAquick PCR purification kit (Qiagen), with elution in 15 µl of Qiagen Elution Buffer (alternatively, phenol-chloroform extraction and ethanol precipitation). Briefly for library construction, ChIP DNA fragments were end-repaired and ligated to indexed sequencing adapters (NEB or Rubicon kit) and amplified with 8–11 rounds of PCR, depending on the DNA amount. Using SPRI beads (Beckman Coulter), small fragments (<200 bp) were removed from the amplified library. The size-selected library was sequenced to 50 bp single-end read lengths on an HiSeq4000 (Vincent J. Coates Genomic Sequencing Laboratory, UC Berkeley).

## Standard ChIP-seq

For standard ChIP-seq, the protocol described by Kitagawa et al was employed (*Kitagawa et al., 2017*).

## Urea chromatin immunoprecipitation (urea ChIP)-western analysis

Urea-purified chromatin prepared as described above (~10 µg of chromatin-DNA) was immunoprecipitated with indicated antibody to produce urea ChIP samples, and the chromatin fragments bound by the antibody were trapped by Protein-A beads. The antibody-bound chromatin fragment-trapped beads were washed with binding buffer/washing buffer/NaCl-TE (total 6 times), and the chromatin fragments were released and isolated from beads (65 °C for 20 min) in 100 µl of 1% SDS, 10 mM Tris, pH7.6, 1 mM EDTA, 100 mM NaCl solution. The isolated chromatin fragments were subjected to reverse-crosslinking by incubation at 65 °C for an additional 6 hr. The released proteins were dissolved in the Laemmli buffer, boiled, and subjected to western blot assay. To compare multiple samples, we confirmed that we used a similar amount of original urea-purified crosslinked chromatin before ChIP by subjecting each chromatin sample to Western blot using anti-H3 antibody. The similar amount of urea-purified crosslinked chromatin was then used for ChIP-western studies.

## Urea 4C-seq

Urea-purified crosslinked chromatin (described above for its preparation) was processed for urea-4C-seq basically as described (*Kohwi-Shigematsu et al., 2012*) with some modifications. For the current experiment, the purified crosslinked chromatin (appears as a transparent pellet at the bottom of a centrifugation tube) after urea centrifugation was briefly sonicated in 100 µl of the *Hind*III digestion buffer with Branson sonifier (20% power, 10 sec, once). Next, *Hind*III was added to digest chromatin (10–20 units/µg DNA) overnight at 37 °C. *Hind*III was heat inactivated at 80 °C for 20 min, ligation was performed in 1.0 ml with 1 X ligation buffer with ligase (600 units) at 16 °C for overnight, followed by reverse-crosslinking at 65 °C for overnight. The DNA was then purified by phenol-chloroform extraction or by spin column (NucleoSpin, Macherey-Nagel). The purified DNAs (1–4 µg) were further digested with *Hae*III (100–125 units) overnight. The *Hind*III-*Hae*III digested DNA fragments were treated with Klenow. The resulting blunt-end *Hind*III-*Hae*III DNA fragments were ligated with our custom double-stranded (ds) Linker DNA with one strand-Biotin modified,

Linker-top strand sequence with Biotin modified (marked by *):
5'-<u>aattc</u>ggtacctctagagat*at*ccgatc<u>gctcgag(Xho</u>I)aagctt(<u>Hind</u>III)–3';
Linker-bottom strand sequence: 5'-aagcttctcgagcgatcggatatctctagaggtaccgaatt-3' as described (*Kohwi-Shigematsu et al., 2012*). The Linker-ligated dsDNA fragments were purified by spin column to remove free Linker. The Linker-ligated dsDNA fragments were further digested with *Xho*I at the linker site (next site from *Hind*III). Single-stranded DNAs were isolated from the *Xho*I digested DNA fragments by coupling to StreptAvidin Dynabeads (Invitrogen, Dynabeads M-280 streptavidin) followed by incubation in 40 µl of 0.15 M NaOH for 10 min at 25 °C, followed by neutralization. Only single-stranded DNA, containing biotin coupled with streptavidin, was retained on the Dynabeads, releasing the non-biotin modified single-stranded DNA (single-strand DNA capture; *Kohwi-Shigematsu et al., 2012*). The released non-biotin modified single-strand DNAs were amplified by PCR (AmpliTaq Gold

system, Applied Biosystems); 10 cycles at 94 C for 30″, 55 °C for 60″ and 72 °C for 60″, followed by 15–20 cycles at 94 °C for 30″, 65 °C for 60″, 72 °C for 60″, using the custom Illumina sequence-bait oligomers and Illumina sequence-linker oligomer for library preparations as described below. The PCR amplified DNAs were purified by high and low size selection by beads (SPRIselect, Beckman Coulter). The purified DNAs were analyzed by Bioanalyzer to confirm their average size of ~300–500 bp with mostly uniform distribution, followed by the Illumina sequencing using custom bait-specific sequence oligomers and Linker-specific barcode oligomer. We used two bait sequences: 5′ gaaggggtagga aagaaagctt 3′ (BUR-1 bait: adjacent to a SATB1-bound BUR, located 1.4 kb from ASE) and 5′ cgact cattcctactgaagctt 3′ (BUR-2-bait: enhancer located in the second intron of B230118H07Rik close to a SATB1-bound BUR), both including a *Hind*III site.

The library was constructed by amplification using either Illumina sequence-BUR-1-bait or Illumina sequence-BUR-2-bait primer with Illumina sequence-linker oligomer. (Capital letters represent Illumina sequence, lower case letters represent bait sequences or the linker sequence)

Illumina sequence-BUR-1-bait oligomer:
5′-AATGATACGGCGACCACCGAGATCTACACTCTTTCCCTACACGAC-gaagggggtaggaaagaa agct

Illumina sequence-BUR-2-bait oligomer:
5′-AATGATACGGCGACCACCGAGATCTACACTCTTTCCCTACACGAC-cgactcattcctactgaagctt

Illumina sequence-linker oligomer: 5′-CAAGCAGAAGACGGCATACGAGAT (barcode; e.g. CGTGAT) GTGACTGGAGTTC-gcgatcggatatctctagaggtaccgaattc

To validate successful experiments, we confirmed that no PCR amplified product was generated by using either the bait oligomer or linker oligomer alone.

For the library construction, Taq-Gold polymerase (NEB) was used for PCR by the following condition: 94 °C for 5–10 min (one cycle), 94 °C for 30 s, 55 °C for 30 s, 72 °C for 60 s (for 5–10 cycles), 94 °C for 30 s, 65 °C for 30 s, 72 °C for 60 s (for 20 cycles), followed by 72 °C for 10 min. PCR products were purified by phenol-chloroform extraction twice, CHCl3 extraction once, followed by EtOH precipitation (Optional; PCR purification by spin column, Qiagen).

The 4 C libraries were sequenced by Illumina-HiSeq4000, using BUR-1-bait sequence and BUR-2-bait sequence as custom sequence oligomers as follows:

BUR-1-bait sequence oligomer: 5′-CTTTCCCTACACGAC-gaagggggtaggaaagaaagctt
BUR-2-bait sequence oligomer: 5′-ACTCTTTCCCTACACGAC-cgactcattcctactgaagctt
Each library sequence was identified with Linker-specific barcode oligomer;

5′-gaattcggtacctctagagatatccgatcgc-GAACTCCAGTCAC. (Capital letters represent Illumina sequence and lower-case letters represent linker sequence). For sequencing, the above custom barcode sequence oligomer was required instead of the common Illumina barcode sequencing oligomer.

## Hi-C

Hi-C was performed as described (*Rao et al., 2014*). Primary thymocyte samples from *Satb1*[+/+]:*Cd4*-Cre and *Satb1*[fl/fl]:*Cd4*-Cre mice were formaldehyde crosslinked at 2% at room temperature for 10 min. Two biological replicates per sample were prepared for Hi-C. Hi-C libraries were prepared using the Arima HiC[+] kit (Arima Genomics) exactly following the associated protocol. Paired-end sequencing was performed by Illumina NovaSeq 6000.

## Statistical analysis
### Significance of overlap of SATB1 with BURS

We used the *bootRanges* function that is part of the *nullranges* bioconductor package (*Mu et al., 2023*; *Huber et al., 2015*). It was used to compute the significance of overlap of two sets of genomic regions (e.g. BUR versus the called peaks for each of the ChIP-seq data sets). This method is based on subsampling genomic regions of a given size from within genomic segments of the same state. Here the mouse genome mm9 was segmented into regions capturing three states based on density of known genes using the segmentDensity function using the CBS method 4 (*Olshen et al., 2004*) and assuming a segment length of 2 Mb. Regions represented blacklisted regions (queried using the terms 'excluderanges' and 'mm9') in the AnnotationHub package were excluded from the analysis. For each comparison [reference regions (known BUR peak calls) versus query regions (SATB1 Urea or

Standard ChIP-seq peak calls)], the observed number of overlapping regions is first computed, 1000 sets of random query region peak calls were sampled given the estimated genome segmentation using blocks of length 300 kb, the overlaps for each of these 1000 random query regions with the reference regions were then computed and finally, the estimated p-value is based on the fraction of 1000 random overlaps greater than or equal to the observed number of overlaps. These results were validated by evaluating the statistical enrichment of SATB1 binding at BURs using the regioneR package (v1.30.0) in R (*Gel et al., 2016*), generating a p value of 0.000999, Z-score of 543.37, and a one-sided test confirmed greater than expected overlap. SATB1-bound genomic regions (satb1_gr) and BURs (bur_gr) were represented as GRanges objects. To determine whether SATB1 binding events occur more frequently at BURs than expected by chance, we performed a permutation test using permTest with 1000 iterations. SATB1 regions were randomly shuffled across the mm9 mouse genome using randomizeRegions, while excluding masked regions such as repeats and assembly gaps via the built-in soft masking available in the BSgenome.Mmusculus.UCSC.mm9 reference. Enrichment was evaluated using numOverlaps to count the number of overlapping SATB1–BUR events in each permutation. A one-sided test was used to assess whether observed overlaps were significantly greater than the randomized distribution.

For genome-wide heatmaps, we quantified SATB1 chromatin occupancy relative to mapped base unpairing regions (BURs). BURs were provided in narrowPeak format, and the SATB1 ChIP-seq peaks were obtained from a processed peak file containing signal intensity. Using the normalizeToMatrix function from the EnrichedHeatmap package (v1.30.0; *Gu et al., 2018*), we computed SATB1 signal intensity in 50 bp bins across a ±3 kb window centered on each BUR. BURs were either plotted as a total genome-wide dataset or stratified into lamina-associated domains (LADs) and inter-LADs (iLADs) based on overlap with DamID-defined LAD intervals using the LADetector script (*Chen et al., 2018*; *Reddy and Wong, 2025*). Normalized signal matrices were used to generate heatmaps of SATB1 enrichment using ComplexHeatmap, with LADs and inter-LADs plotted separately. Line plots above the heatmaps show the average SATB1 signal across all BURs in each category. Color scaling was applied using colorRamp2 from the circlize package (v0.4.15; *Gu et al., 2014*). All visualizations were constructed using ComplexHeatmap (v2.18.0), with rasterization enabled for performance.

To assess the alignment of BURs to SATB1-bound loci, we performed the inverse analysis by quantifying BUR signal intensity around SATB1 binding peaks. SATB1 peaks were split into LAD- and iLAD-associated groups using genomic overlap with LAD intervals. BURs were used as a quantitative signal (score column from the narrowPeak file), and enrichment around each SATB1 site was computed in 50 bp bins across a ±3 kb window using normalizeToMatrix function from the EnrichedHeatmap R package (v1.30.0). The resulting matrices were visualized as heatmaps and line plots of average BUR signal genome-wide or in each domain context (LAD vs iLAD). All visualizations were constructed using ComplexHeatmap (v2.18.0).

## Hi-C data processing

Hi-C datasets were analyzed using the HiC-bench platform (*Lazaris et al., 2017*). The application of this method is described in detail (*Kloetgen et al., 2020*). Briefly, data were aligned against the mouse reference genome (mm10) by bowtie2 (version 2.3.1) (*Langmead and Salzberg, 2012*) with mostly default parameters (specific settings: `--very-sensitive-local --local`). For Hi-C, aligned reads were filtered by the GenomicTools (*Tsirigos et al., 2012*) tools-hic filter command integrated in HiC-bench. Interaction matrices for each chromosome separately were created by the HiC-bench platform at a 10 kb resolution. The Hi-C filtered contact matrices were normalized using 'iterative correction and eigenvector decomposition' (ICE) algorithm (*Imakaev et al., 2012*) built into HiC-bench. To account for variances of read counts of more distant loci, distance normalization for each matrix was performed as recently described (*Gong et al., 2018*). TADs were called using the algorithm developed within HiC-bench (*Lazaris et al., 2017*) at 10 kb bin resolution with the insulating window of 100 kb. For visualization of Hi-C data, we generated HiC files using the Juicer (*Durand et al., 2016*).

## Quality control

The total numbers of reads per biological replicate for each genotype ranged from 200 to 230 million reads. The percentage of reads aligned was >98% in all samples. The percentage of accepted reads ('ds-accepted-intra' and 'ds-accepted-inter') was in the range of 50–55.5%. The absolute number of

accepted intra-chromosomal read pairs was ~90–100 million reads for each sample. We analyzed data from individual samples as well as combined data from two replicates (~200 million reads) for each genotype.

A scaling plot was performed by aggregating the counts per 10 kb distance bins. Values were cpm-normalized by dividing by the total number of valid intra-chromosomal pairs.

## Compartments

Compartment analysis was performed using the Homer pipeline (v4.6; *Heinz et al., 2010*). Homer performs a principal component analysis of the normalized interaction matrices and uses the PCA1 component (cis-Eigenvector 1) to predict regions of transcriptionally active (A compartments) and transcriptionally inactive chromatin (B compartments). Homer works under the assumption that gene-rich regions with active chromatin marks have similar PC1 values, while gene deserts show the opposite (http://homer.ucsd.edu/homer/interactions/HiCpca.html). HiC filtered matrices were given as input to Homer together with a list of house-keeping genes for compartment prediction. The house-keeping genes coordinates are used by Homer as prior information of active regions. We generated density plots to compare the cis-eigenvector 1 values of the knock-out and wild-type samples with 10 kb genomic bins. Pearson correlation coefficients were calculated ('Cor' function in R) between the counts for every pair of 10 kb bins on chromosome 2. Heatmaps representing the Pearson correlation of interactions of chromosome 2 for WT and KO thymocytes are displayed by juice box (500 kb resolution).

## urea ChIP-seq analysis

Sequenced reads were quality-checked using FastQC prior to downstream analysis. Unaligned reads were mapped to the mouse reference genome (NCBI37/mm9) using bowtie2 (version 2.3.1) (*Langmead and Salzberg, 2012*) allowing for three mismatches in the seed region and only unique matches. Candidate binding sites were called using MACS2 (v2.2.7) (*Zhang et al., 2008*) using a q-value threshold of 0.01. Coverage tracks are presented in RPM (read pileups per million reads) and utilities from UCSC were used for creating browser tracks (*Kent et al., 2010*). To determine the intersection of peaks across ChIP-seq samples, deepTools (*Ramírez et al., 2016*) was used.

## urea 4C-seq analysis

Reads were mapped with Bowtie, then assigned to *HindIII* sites with the same utility tools used in *Ben Zouari et al., 2019*. These same tools also normalized the counts to per million intrachromosomal reads and removed reads to restriction fragments comprising more than 2% of the total reads (namely the contiguous adjacent restriction fragment corresponding to undigested chromatin during the urea 4 C, and a rare (<5) numbers of fragments with extreme PCR duplications). The generated bedGraph files and these were converted to bigWig format using the bedGraphToBigWig program for direct visualization of the peaks on the UCSC browser.

## Analysis of SATB1-bound sites and other genomic features

LADs were identified as genomic intervals using the previously described LADetector algorithm (https://github.com/thereddylab/pyLAD; *Sauria and Luperchio, 2019*) or sequencing data (https://github.com/thereddylab/LADetector; *Luperchio, 2018*) (*Harr et al., 2015*; *Zullo et al., 2012*). SATB1 urea-ChIP data from thymocytes were tested for statistically significant overlap of LADs using the GenometriCorr package (*Favorov et al., 2012*). The GenometriCorr package applies multiple spatial tests of independence including a relative distance test (i.e. if two elements such as LADs and SatB1 sites are spaced more often than expected at a certain relative distance from each other), projection test (i.e. testing for significant overlap between these two positions assuming they represent single points), and the Jaccard test (testing for correlation of these two positions assuming that they are not points but instead occupy some interval of the genome).

To generate genome-wide line plots of averages of intensities (relative score) of DamID and ChIP-seq peak data were generated over inter-LAD (10% of the size of the adjacent LAD)/border

flanked LADs scaled to the same relative size using the Genomation R package (*Akalin et al., 2015*). The average intensities were then normalized to fit a scale of –1–1 and plotted over the LADs (normalized relative score). To plot genomic features in the vicinity of enhancers, H3K4me1 peaks outside of LADs were filtered by subtracting away peaks that intersected LADs using the BEDTools (*Quinlan and Hall, 2010*). To plot genomic features in the vicinity of DiPS in LADs, regions within LADs and DiPS were identified by LADetector. DiPS were then mapped as intervals, and the method described above was applied to DiPS versus SATB1, CTCF, and chromatin modifications. Genome-wide averages of the respective ChIP-seq/DamID signal data were then generated within a ±3 kb window centered at H3K4me1 peaks, using Genomation R package and plotted on a vertical scale ranging from –1 to 1.

## Acknowledgements

We thank Dr. Aristotelis Tsirigos, Ziyan Lin, and Javier Rodriguez Hernaez at New York University School of Medicine for Hi-C and intersection analyses, Dr. Anthony Schmitt from Arima Genomics for help with Hi-C experiments. We also thank Dr. Elphege Nora for reviewing Hi-C results and providing critical and productive comments on the manuscript and Ms. Bao Ho for providing technical assistance. We thank Dr. Minoree Kohwi for critical reviewing of the manuscript and Dr. Peter Krijger for kindly testing our samples and Dr. Elzo de Wit for discussion on the 4C-seq analysis. National Institutes of Health grant R31CA39681 (TKS) National Institutes of Health grant R01ES023854 (TKS) National Institutes of Health grant R01GM132427 (KLR) National Institutes of Health grant R01AR071727 (VAB and TKS) National Institutes of Health grant R01 AR047364 (C-MC and TKS).

## Additional information

### Funding

| Funder | Grant reference number | Author |
| --- | --- | --- |
| National Institutes of Health | R31CA39581 | Terumi Kohwi-Shigematsu |
| National Institutes of Health | R01ES023854 | Terumi Kohwi-Shigematsu |
| National Institutes of Health | R01AR071727 | Vladimir A Botchkarev Terumi Kohwi-Shigematsu |
| National Institutes of Health | AR047364 | Cheng-Ming Chuong Terumi Kohwi-Shigematsu |
| National Institutes of Health | R01GM132427 | Karen L Reddy |

The funders had no role in study design, data collection and interpretation, or the decision to submit the work for publication.

### Author contributions

Yoshinori Kohwi, Conceptualization, Resources, Data curation, Formal analysis, Supervision, Validation, Investigation, Methodology, Writing – original draft, Project administration, Writing – review and editing, Devised and optimized the urea ChIP-seq method; Xianrong Wong, Data curation, Software, Formal analysis, Validation, Investigation, Analyzed the relative positions of LADs, BURs, and SATB1-binding sites; Mari Grange, Hunter W Richards, Data curation, Formal analysis; Thomas Sexton, Data curation, Software, Formal analysis, Validation, Methodology, Processed and validated urea 4C-seq data; Yohko Kitagawa, Shimon Sakaguchi, Resources, Data curation, Writing – review and editing; Ya-Chen Liang, Formal analysis; Cheng-Ming Chuong, Funding acquisition, Writing – review and editing; Vladimir A Botchkarev, Resources, Funding acquisition; Ichiro Taniuchi, Resources, Writing – review and editing; Karen L Reddy, Formal analysis, Supervision, Funding acquisition, Validation, Investigation, Methodology, Writing – original draft, Writing – review and editing, Supervised the work on LADs in this study; Terumi Kohwi-Shigematsu, Conceptualization, Resources, Data curation, Formal analysis, Supervision, Funding acquisition, Validation, Investigation, Visualization, Methodology,

Writing – original draft, Project administration, Writing – review and editing, Designed all experiments and supervised the project

### Author ORCIDs
Yoshinori Kohwi https://orcid.org/0000-0002-5443-0909
Xianrong Wong https://orcid.org/0000-0002-1995-7043
Thomas Sexton https://orcid.org/0000-0002-7824-1846
Hunter W Richards https://orcid.org/0000-0001-9913-089X
Shimon Sakaguchi https://orcid.org/0000-0001-8265-6094
Ya-Chen Liang https://orcid.org/0000-0001-7813-8921
Cheng-Ming Chuong https://orcid.org/0000-0001-9673-3994
Vladimir A Botchkarev https://orcid.org/0000-0002-5212-5353
Ichiro Taniuchi https://orcid.org/0000-0002-9853-9068
Karen L Reddy https://orcid.org/0000-0003-4548-5999
Terumi Kohwi-Shigematsu https://orcid.org/0000-0002-5988-0669

### Ethics
This study was performed in strict accordance with the Guides for the Care and Use of Laboratory Animals of the National Institutes of Health. All the animals were handled according to approve institutional animal care and use committee (IACUC) protocols AN192808-01C and AN206117-00 of University of California, San Francisco.

Reviewer #1 (Public review): https://doi.org/10.7554/eLife.105915.3.sa1
Reviewer #2 (Public review): https://doi.org/10.7554/eLife.105915.3.sa2
Author response https://doi.org/10.7554/eLife.105915.3.sa3

---

## Additional files

### Supplementary files
MDAR checklist

Supplementary file 1. Intersection analysis of ChIP-seq samples.

Supplementary file 2. ChIP-seq sample identification and peak number.

Supplementary file 3. Intersection analysis of WT and KO ChIP-seq samples.

Supplementary file 4. WT and KO ChIP-seq sample identification and total peak number.

### Data availability
All sequencing data for ChIP-seq and Hi-C in this study have been deposited in Gene Expression Omnibus (GEO) database and are available under the accession number GSE191146. We also used publicly available ChIP-seq data: H3K27ac and H3Kme1 ChIP-seq data in wild-type and *Satb1fl/fl:Cd4-Cre* immature CD4SP (imCD4SP) and thymic Treg cells are available in the NCBI SRA database (SRA accession number DRP003376; *Kitagawa et al., 2017*), and JARID2, SUZ12, EZH2 ChIP-seq data are available in the GEO database (accession number GSE18776; *Peng et al., 2009*). We also used published RNA-seq data for wild-type and *Satb1fl/fl:Cd4-Cre* thymocytes available in the GEO database (accession number GSE173476; *Zelenka et al., 2022*).

The following dataset was generated:

| Author(s) | Year | Dataset title | Dataset URL | Database and Identifier |
|---|---|---|---|---|
| Kohwi Y, Wong X, Grange M, Sexton T, Richards HW, Liang Y, Chuong C, Kitagawa Y, Sakaguchi S, Botchkarev VA, Taniuchi I, Reddy KL, Kohwi-Shigematsu T | 2021 | Deeply hidden genome organization mediated by base-unpairing regions (BURs) directly bound by SATB1 linked to transcription | https://www.ncbi.nlm.nih.gov/geo/query/acc.cgi?acc=GSE191146 | NCBI Gene Expression Omnibus, GSE191146 |

The following previously published datasets were used:

| Author(s) | Year | Dataset title | Dataset URL | Database and Identifier |
|---|---|---|---|---|
| Zelenka T, Klonizakis A, Tsoukatou D, Papamatheakis DA | 2022 | The 3D enhancer network of the developing T cell genome is shaped by SATB1 | https://www.ncbi.nlm.nih.gov/geo/query/acc.cgi?acc=GSE173476 | NCBI Gene Expression Omnibus, GSE173476 |
| Yohko K, Naganari O, Yujiro K, Alexis V, Keiji H, Ryoji K, Daisuke M, Shota N, Motonari K, Ichiro T, Terumi KS, Shimon S | 2017 | Guidance of regulatory T cell development by Satb1-dependent super-enhancer establishment | https://www.ncbi.nlm.nih.gov/sra/?term=DRP003376 | NCBI Sequence Read Archive, DRP003376 |
| Peng JC, Valouev A, Swigut T, Zhang J, Zhao J, Sidow Y, Wysocka A | 2009 | Jarid2/Jumonji coordinates control of PRC2 enzymatic activity and target gene occupancy in pluripotent cells | https://www.ncbi.nlm.nih.gov/geo/query/acc.cgi?acc=GSE18776 | NCBI Gene Expression Omnibus, GSE18776 |

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
