## [Editor Report · eLife Assessment]

This is a very **important** study in which the authors have modified ChIP-seq and 4C-seq with a urea step, which drastically changes the pattern of chromatin interactions observed for SATB1, but not other proteins (including CTCF). The study highlights that the urea protocols provide a complementary view of protein-chromatin interactions for some proteins, which can uncover previously hidden, functionally significant layers of chromatin organization. If applied more widely, these protocols may significantly further our understanding of chromatin organization. The study's findings are supported by a wealth of controls, making the evidence **compelling**.

---

## [Referee Report · Reviewer #1 (Public review)]

Summary:

The nuclear protein SATB1 was originally identified as a protein of the 'nuclear matrix', an aggregate of nuclear components that arose upon extracting nuclei with high salt. While the protein was assumed to have a global function in chromatin organization, it has subsequently been linked to a variety of pathological conditions, notably cancer. The mapping of the factor by conventional ChIP procedures showed strong enrichment in active, accessible chromatin, suggesting a direct role in gene regulation, perhaps in enhancer-promoter communication. These findings did not explain why SATB1-chromatin interaction resisted the 2 M salt extraction during early biochemical fractionation of nuclei.

The authors, who have studied SATB1 for many years, now developed an unusual variation of the ChIP procedure, in which they purify crosslinked chromatin by centrifugation through 8 M urea. Remarkably, while they lose all previously mapped signals for SATB1 in active chromatin, they now gain many binding events in silent regions of the genome, represented by lamin-associated domains (LADs).

SATB1 had previously been shown by the authors and others to bind to DNA with special properties, termed BUR for 'base-unpairing regions'. BURs are AT-rich and apparently enriched in equally AT-rich LADs. The 'urea-ChIP' pattern is essentially complementary to the classical ChIP pattern. The authors now speculate that the previously known SATB1 binding pattern determined by standard ChIP, which does not overlap BURs particularly well, is due to indirect chromatin binding, whereas they consider the urea-ChIP profile, which fits better to the BUR distribution on the chromosome, to be due to direct binding.

Building on the success with urea-ChIP the authors adapted the 4C-procedure of chromosome conformation mapping to work with urea-purified chromatin. The data suggest a model according to which BUR-bound SATB1 mediates long-distance interaction between active loci and some kind of scaffold structure formed by SATB1. Because cell type-specific differences are observed, they suggest that the SATB1 interactions are functionally relevant.

Strengths:

Given the unusual findings of essentially mutually exclusive 'standard ChIP' and 'urea-ChIP' profiles, the authors conducted many appropriate controls. They showed that all SATB1 peaks in urea-ChIP and 96% of peaks in standard-ChIP represent true signals, as they are not observed in a SATB1 knockout cell line. They also show that the urea-ChIP and standard ChIP yield similar profiles for CTCF and polycomb complex subunits. The data appear reproducible judged by at least two replicates and triangulation. The SATB1 KO cells provide a nice control for the specificity of signals, including those that arise from their elaborately modified 4C protocol.

In their revised manuscript the authors provide relevant background information concerning the effect of urea on the denaturation of macromolecules. Importantly, they argue convincingly that urea does not denature DNA under their conditions.

Weaknesses:

Despite the authors' efforts to explain their findings along with a lot of background information, some readers may be left confused due to the complexity of the system. BURs are found enriched in LADs, but are also present in active chromatin. SATB1 binds a subset of BURs, but the reason for discrimination remains unclear. SATB1 appears to bind chromatin in at least two modes with differing diffusion properties and exactly how this relates to the indirect and direct chromatin binding modes is mechanistically unclear.

The authors resort to the term 'SATB1-enriched subnuclear structure' to describe the profile gained through denaturing ChIP, thus avoiding strong statements about involvement of known nuclear structures (such as LADs or heterochromatin) and about functional implications.

The authors acknowledge a potential for RNA to be involved in modulating SATB1 interactions with chromatin, but leave this for future investigation.

Comment on revised version:

The authors revised their manuscript to my satisfaction.

---

## [Referee Report · Reviewer #2 (Public review)]

Summary:

This study describes the key observation that SATB1 binds directly to so-called BUR elements. This is in contrast to several other reports describing SATB1 binding to promoters and enhancers. This discrepancy is explained by the authors to depend on the features of the ChIP technique being used. Urea-ChIP, innovated by the authors, strips off protein-protein interactions that compound conventional ChIP methods. The authors convincingly make the case that SATB1 and a key genome organiser, CTCF, largely bind different sites, as particularly evident in Figure 2A. In contrast, standard ChIP shows considerable overlap between their sites (Figure 2-figure supplement 1). The report documents convincingly that SATB1 partitions the genome independent of so-called TADs to influence expression patterns. SATB1 controls long-range interactions in thymocytes, and knock down of SATB1 does not affect the TAD patterns.

Strengths:

A new and innovative adaptation of ChIP-seq (urea ChIP-seq) has enabled the authors to successfully question existing data on the patterns of SATB1 binding to the genome. The authors provide a wealth of data to reinforce their claims. This report thus rectifies misconceptions about SATB1 function, which is particularly important given its role in metastasising cancer cells.

Weaknesses:

None

---

## [Author Response]

The following is the authors’ response to the original reviews

**Reviewer #1:**
(1) 8 molar urea not only denatures proteins but also denatures DNA. Obviously, this does not affect the ChIP, since antibodies often recognize small linear epitopes and the proteins are crosslinked. However, under high urea conditions the BUR elements should be rendered single-stranded, and one wonders whether this has any effect on the procedure. The authors should alert the reader of these circumstances.

Thank you for raising this important question about the effects of 8M urea. We have added a brief paragraph explaining this point in the revised manuscript. Despite common misconceptions, 8M urea by itself does not actively convert double-stranded DNA to single-stranded DNA. For this conversion to occur, a heat denaturation step is required. Once DNA is heat-denatured to become single-stranded, urea can maintain this configuration. This is why the addition of 8M urea to acrylamide gel electrophoresis is a standard method for analyzing single-stranded oligonucleotides, but the DNA must first be denatured by heat (Summer et al., J. Vis. Exp. (32), e1485, DOI : 10.3791/1485). This is clearly described in published work comparing the status of DNA with and without heat treatment in an 8M urea-containing buffer Hegedus et al., Nucl.Acids Res. 2009 (doi:10.1093/nar/gkp539).

We have additional evidence supporting this conclusion in the context of our urea ultracentrifugation experiment. Both crosslinked and un-crosslinked genomic DNA purified by 8M urea centrifugation can be digested with restriction enzymes, which indicates that the DNA remains double-stranded. For instance, we previously published SATB1 ChIP-3C results using Sau3A-digested DNA after urea purification. In the current paper, we used *Hind*III to digest urea-purified DNA for urea 4C-seq. The BUR reference map can also be generated after restriction digestion of urea-purified DNA and isolating and sequencing SATB1-bound restriction fragments in *vitro*. If genomic DNA were denatured by 8M urea ultracentrifugation, we would not have been able to digest it with restriction enzymes to obtain these results.

We have now added a sentence noting that SATB1 is a double-stranded DNA-binding protein that does not bind to single-stranded DNA, as we have previously shown (Dickinson et al., 1992, Ref 32).

(2) An important conclusion is that urea-ChIP reveals direct DNA binding events, whereas standard ChIP shows indirect binding (which is stripped off by urea). I do not see any evidence for direct binding. At low resolution, predicted BUR elements are enriched in domains where SATB-1 is mapped by urea-ChIP. A statement like 'In a zoomed-in view, covering a 430 kb region, SATB1 sites identified from urea ChIP-seq precisely coincided with BUR peaks' is certainly not correct: most BUR peaks do not show significant SATB-1 binding. The randomly chosen regions shown in Figure 4 – Supplement 1 show how poor the overlap of SATB-1 and BURs is; indeed, they show that SATB-1 binds DNA mostly at non-BUR sites. I see Figure 2D, but such cumulative plots can be highly biased by very few cases. I suggest showing these data in heat maps instead.

We believe there may be some confusion regarding the interpretation of our figures. Looking at Track 3 (BUR reference map, RED peaks) and urea SATB1 Tracks 4 and 5 (replicas from two independent experiments) in Figure 2B, the SATB1 peaks detected by urea ChIP-seq do indeed coincide with BUR peaks. In the revised manuscript, we have provided a further ‘zoomed-in’ view to better illustrate this point and also provided the underlying BUR sequence from one of these SATB1-bound regions (Figure 2—supplement figure 1).

It is true that many more BURs exist than SATB1-bound BURs, especially in gene-poor regions where BURs are clustered. However, from the perspective of SATB1-bound peaks, the majority of these coincide with BURs, as shown by both deepTools analyses and new heatmap, as suggested (Figure 2E, and Figure 7—supplement figure 3).

The results from our genome-wide quantitative analyses using deepTools to compare peaks from urea SATB1 ChIP-seq data and the BUR reference map shown in Supplementary Tables 1 and 2 are consistent with the heatmap analyses.

We must apologize for an error in the scaling of the y-axis in Figure 4-supplement figure 1 that likely contributed to some confusion. We have corrected our mistake in the revised manuscript. As we were preparing our figures, when placed in the figure and axes relabeled for legibility, the BUR reference peaks were mislabeled on their y-axis. In the figure the peaks were erroneously labeled on a scale of 0.1-1 read counts/million reads, but the data shown is actually scaled at 0.1 to 2 read counts per million reads. Unfortunately, we did not realize this error and, using the figure as a guide for scaling, provided urea SATB1 ChIP-seq peaks at a scale of 0.1-1 read counts/million reads to match the mislabeled BUR reference track. This had the effect of reducing the signal/noise in the SATB1 ChIP-seq data (Figure 1). We have now standardized the y-axis for fair comparison using a scaling of the y-axis at 0.1-2 for all tracks. This will more clearly show that there are indeed more BUR peaks than SATB1-bound sites, consistent with our quantitative analysis.

We hope that these clarifications as well as the added heatmaps and binding site example allay the concerns about the specificity and overlap of SATB1 binding on BURS.

(3) In Figure 6C 'peaks' are compared. However, looking at Figure 4 - Supplement 1 again it is clear that peak calling can yield a misleading impression. Figure 6D suggests that there are more BURs than SATB-1 peaks but this is not true from looking at the browser.

We thank the reviewer for this observation. As noted in our response to point 2 above, the inconsistent y-axis scaling in Figure 4-supplement figure 1 created a misleading impression, which we have corrected in the revised manuscript. When properly displayed with consistent y-axis scaling, the browser view aligns with our quantitative data showing that there are indeed many more BURs than SATB1-bound sites. As mentioned under 2 above, we have performed genome-wide quantitative analysis by deepTools (Supplementary Tables 1 and 2) to confirm the results shown by bar graphs in Fig. 6C, 6D and Fig. 2D.

In Figure 6C, the bars show the percentage of SATB1-bound peaks in each cell type (denominator) that overlap with confirmed BUR sites in the BUR reference map (numerator). In Figure 6D, we show the percentage of total BUR sites in the BUR reference map (denominator) that are bound by SATB1 from urea ChIP-seq (numerator). To avoid any confusion, we have added brief subtitles to Figures 6C and 6D in the revised manuscript.

(4) An important conclusion is that urea-ChIP reveals direct DNA binding events, whereas standard ChIP shows indirect binding (which is stripped off by urea). I do not yet see any evidence for direct binding. It cannot be excluded that the binding is RNA-mediated. The authors mention in passing that urea-ChIP material still contains (specific!) RNA. Given that this is a new procedure, the authors should document the RNA content of urea-ChIP and RNase-treat their samples prior to ChIP to monitor an RNA contribution.

Thank you for raising this important point. The direct binding of SATB1 to BURs is well-established in our previous work. Indeed, this was the main motivation to explore the reason for the lack of evidence for genome-wide SATB1 binding to BURs in the DNA-binding profile by standard ChIP-seq. This has been a major point of confusion for us for many years.

SATB1 was originally identified through a search for mammalian proteins that could recognize BURs specifically and not just any A+T-rich sequence. The *Satb1* gene was originally cloned by an expression cDNA library and encoded SATB1 protein bound the BUR probe but not a mutated AT-rich BUR (control) probe. Subsequent experiments confirmed that SATB1 specifically binds to many BURs without requiring additional factors. Furthermore, SATB1 recognizes BURs by binding in the minor groove of double-stranded DNA, presumably recognizing the altered phosphate backbone structure of BUR DNA, rather than accessing nucleotide bases (Dickinson et al, 1992).

We do agree with the reviewer, however, that there is a possibility that RNA can redirect SATB1 to different subsets of BURs and/or to interact indirectly with different regulatory regions depending on cell type or developmental stage. Although urea ultracentrifugation clearly separates most RNA (found in the middle region of the tube) from genomic DNA (pelleted at the bottom) (de Belle et al., 1998), upon crosslinking cells, a small quantity of RNA is found co-pelleted with DNA (our recent unpublished results). This RNA, tightly associated with crosslinked chromatin, may have some impact on SATB1 function.

Based on our preliminary data, we are currently planning to study the impact of RNA using RNase A as well as by targeting specific RNAs employing an anti-sense approach. We believe that thoroughly addressing the impact of RNA warrants a full paper, including the potential roles of specific non-coding RNAs in SATB1 function, and thus is beyond the scope of the current paper. However, we have now added discussion of this important point in the manuscript.

(5) An important aspect of the model is that SATB1 tethers active genes to inactive LADs. However, in the 4C experiment the BUR elements used to anchor the looping are both in the accessible, active chromatin domain. If the authors want to maintain their statement, they must show a 4C result that connects the 2 distinct domains and transverses A/B domain boundaries. Currently, the data only show a looping within accessible chromatin.

We appreciate REVIEWER 1 for bringing up the important point that our model could potentially be interpreted as “SATB1 tethers active genes to inactive LADs.” Since we describe that BURs are enriched in LADs and that SATB1 binds a subset of BURs, readers may assume that we aim to demonstrate, through urea 4C-seq, that SATB1 tethers active genes to transcriptionally-inactive LADs (via BURs). However, this is not our intention in the model (Figure 8). In the experiment we designed for our present study, we selected BUR-1 and BUR-2 as viewpoints from a non-LAD gene-rich region (inter-LAD). Because these BURs are bound by SATB1, it indicates that these BURs are part of the “hard-to-access” SATB1-rich subnuclear structure, which resists extraction, in contrast to accessible chromatin. Thus, we illustrate in the model that BURs anchored to the SATB1-rich nuclear substructure make contact with accessible chromatin over long distances in a SATB1-dependent manner. Therefore, we do not intend to conclude that SATB1 mediates interactions between LADs and inter-LADs (accessible chromatin) from our current study: this would be a topic for future research. In the original model in the submitted manuscript, we used the terms “inaccessible” and “accessible.” In the revised version, we clarified this in the model by changing “inaccessible” to “SATB1-rich subnuclear structure” and carefully revised the text in the Figure 8 legend to clarify the model.

At this time, we do not know exactly how LADs and SATB1 nuclear architecture are related spatially and functionally. While LADs are mapped as genomic domains in proximity to Lamin B1 by LaminB1-DamID, BURs are mapped at ~300-500 bp resolution by urea ChIP-seq. To gain further insight into this important question, a large body of DNA-FISH and immunoDNA-FISH experiments will be required, comparing different cell types to see whether and how specific BURs move between LADs and SATB1 nuclear architecture. Such experiments may benefit from testing the Gabrg1 and Gabra2 loci, where many BURs are anchored to SATB1 in neurons but not in thymocytes, for instance. This is included in Discussion in the revised manuscript.

Regarding the reviewer's second point about showing more extended domains for 4C interactions, we would like to highlight that Figure 5—supplement figure 3 in our submitted manuscript addresses this concern. This figure shows that BUR-interactions extend to multiple gene-rich regions across intervening gene-poor regions. Interestingly, BUR-1 and BUR-2 interactions skip a transcriptionally silent gene-rich region containing olfactory receptor genes but interact with subsequent gene-rich regions containing active genes. These data demonstrate that BUR-interactions do indeed traverse A- and B-compartment boundaries. In the revised manuscript (in Figure 5—supplement figure 3), we newly added a Lamin B1-DamID (thymocyte) track. Comparing with LADs, BUR-1 interactions occur mostly in non-LAD regions. Some minor overlap with LADs was detected in high resolution views (not shown). Future experiments testing BUR viewpoints that reside within LADs are required to assess whether SATB1 mediates interactions between B and A compartments.

(6) The description of the urea-co-immunoprecipitation experiment (Figure 3C) could be improved to make it unequivocally clear that co-binding to chromatin is tested, not protein-protein interaction (which is destroyed by urea).

Thank you for this helpful suggestion. We have revised the text in the manuscript by stating “Distinct from protein-protein co-immunoprecipitation (co-IP) using whole cell or nuclear extracts, we examined the direct co-binding status on chromatin in vivo of SATB1 and CTCF or cohesin by urea ChIP-Western”*.*

**Reviewer #2:**
(1) Since SATB1 has been described to interact with beta-catenin, I wonder if the authors have looked at TCF4/TCF7l2 binding patterns and their potential overlap with SATB1 binding patterns. This might appear a trivial request. However, uncontrolled WNT signalling is a major feature of cancer undergoing metastasis - a process that the authors have earlier associated with unscheduled SATB1 expression in triple-negative breast cancer.

We thank the reviewer for highlighting this important point about the potential relationship between SATB1 and TCF4/TCF7l2 binding patterns. Based on published observations with other factors (Rad21, CTCF, BRG1, RUNX) that show substantial overlap with SATB1 in standard ChIP-seq peaks(Kakugawa et al., Cell Rep 19, 1176-1188 (2017). DOI: 10.1016/j.celrep.2017.04.038. Poterlowicz et al., PLoS Genet, 2017 DOI: 10.1371/journal.pgen.1006966), we would anticipate that TCF4 might also show significant overlap with SATB1. An important question is whether the DNA binding profile of TCF4 depends on SATB1.

We have not yet generated ChIP-seq data for TCF4 in the presence and absence of SATB1, but we agree that such experiments could provide important insights into cancer progression as well as brain function. This represents an interesting direction for future work. We have added this point in our discussion based on your kind suggestion.

(2) The CTCF sizes indicated in the western blot analyses of Figures 3C and Figure 3 - supplement figure 2 do not display the normal size, which is around 130 kDa. Either the issue is erroneous marking or a so-called salt effect to slow the migration in the gel. Alternatively, it reflects a slower migrating form of CTCF generated by for example PARylation (by PARP1) that is known to approach 180 kDa. It would be useful if the authors could clarify this minor issue.

We appreciate the reviewer pointing out this discrepancy. As the reviewer correctly noted, CTCF can appear at a higher molecular weight due to post-translational modifications such as PARylation and O-GlcNAcylation, which alter its migration during electrophoresis.

Upon re-examination of our raw data for Figure 3—supplement figure 2A, we discovered that the marker lane for the CTCF panel was broken, and the 150kDa band was erroneously assigned. This led to the 150kDa marker being placed below the CTCF migration position, which is clearly an error. We thank the reviewer for bringing this to our attention.

We have checked our other data and consistently observe CTCF migrating below the 150kDa band, similar to the pattern shown on the Abcam website for the antibody we used (ab128873) (Figure 2). For Figure 3-supplement figure 2, we will use a marker lane from a parallel gel with identical composition and run time to correctly indicate the molecular weight. We havealso corrected the marker position in Figure 3C.

**Reviewing Editor (Recommendations for the authors):**
(1) The introduction states that urea ChIP-seq is "unbiased", which is difficult to unambiguously determine and therefore might be an overstatement. Maybe the authors could consider rephrasing.

We agree with the reviewer's assessment and have rephrased our description of the urea ChIP-seq method to avoid using the term "unbiased."

(2) The authors propose that in standard ChIP, most SATB1 is in the insoluble fraction. This seems easy to test and demonstrating it may help to further clarify the differences between the protocols.

We appreciate this suggestion and would like to clarify our description. What we stated in the manuscript was:

"We envision that SATB1 bound to inaccessible nuclear regions may be lost in the insoluble fraction."

This refers specifically to a subpopulation of SATB1 that is bound to the high-salt extraction-resistant nuclear substructure, not to the total SATB1 protein. We also noted elsewhere in the manuscript that:

"SATB1 proteins are found in high salt-resistant fraction as well as salt-extracted fraction (40). Thus, it is possible that soluble SATB1 may associate with open chromatin."

Our unpublished results show that SATB1 proteins exist in at least two distinct forms based on protein mobility: SATB1 with high mobility and another with very low or no mobility. While we have identified the SATB1 domain responsible for each of these distinct mobility patterns, we have not yet identified biochemical differences that would allow us to distinguish them conclusively. Therefore, an experiment to test the distribution of SATB1 in soluble versus insoluble fractions would show SATB1 in both fractions but would not necessarily provide information about the functional significance of these different populations. We believe this is an important area for future research and are working to develop tools to specifically distinguish and characterize SATB1 in the soluble versus insoluble fractions.